# A global analysis of the dry-dynamic forcing during cyclone growth and propagation

Philippe Besson[1], Luise J. Fischer[1,2], Sebastian Schemm[1], and Michael Sprenger[1]

[1]Institute for Atmospheric and Climate Science, ETH Zürich, Switzerland
[2]Institute for Environmental Decisions, ETH Zürich, Switzerland

**Correspondence:** michael.sprenger@env.ethz.ch

**Abstract.** Mechanisms driving the intensification and propagation direction of extratropical cyclones are an active field of research. Dry-dynamic forcing factors have been established as fundamental drivers of the deepening and propagation of extratropical cyclones, but their climatological interplay, geographical distribution and relatedness to the observed cyclone deepening and propagation direction remains unknown. This study considers two key dry-dynamic forcing factors, the Eady Growth Rate (EGR) and the upper-level induced quasi-geostrophic lifting ($QG\omega$), and relates them to the surface deepening rates and the propagation direction during the cyclones' growth phase. To this aim, a feature-based cyclone tracking is used and the forcing environment is climatologically analyzed based on ERA-Interim data. The interplay is visualized by means of a forcing histogram, which allows one to identify different combinations of EGR and $QG\omega$ and their combined influence on the cyclone deepening (12-hour sea-level pressure change) and propagation direction. The key results of the study are: (i) The geographical locations of four different forcing categories, corresponding to cyclone growth in environments characterized by low $QG\omega$ and low EGR (Q↓E↓), low $QG\omega$ but high EGR (Q↓E↑), high $QG\omega$ and low EGR (Q↑E↓) and high $QG\omega$ and EGR (Q↑E↑), displays distinct hot spots with only mild overlaps. For instance, cyclone growth in a Q↑E↑ forcing environment is found in the entrance regions of the North Pacific and Atlantic storm tracks. Category Q↓E↑ is typical found over continental North America, along the southern tip of Greenland, over parts of East Asia and the western North Pacific. In contrast, category Q↑E↓ dominates the subtropics; (ii) the four categories are associated with different stages of the cyclones' growth phase: large EGR forcing occurs typically earlier, during the growth phase at genesis, while large $QG\omega$ forcing attains its maximum amplitude later towards maturity; (iii) poleward cyclone propagation is strongest over the North Pacific and North Atlantic, and the poleward propagation tendency becomes more pronounced as the deepening rate gets larger; zonal, or even equatorward propagation, on the other hand, is characteristic for cyclones developing in the lee of mountain ranges, e.g., to the lee of the Rocky Mountains. The exact location of maximum $QG\omega$ forcing relative to the surface cyclone center is found to be a good indicator for the direction of propagation, while no information on the propagation direction can be inferred from the EGR. Ultimately, the strength of the poleward propagation and of the deepening are inherently connected and the two dry-dynamic forcing factors, which allow cyclone development in distinct environments to effectively be identified.

# 1 Introduction

Extratropical cyclones tend to grow and propagate in narrow latitudinal bands known as the storm tracks (Jones and Simmonds, 1993; Chang et al., 2002; Hoskins and Hodges, 2002; Wernli and Schwierz, 2006), but they also occur frequently outside of the main oceanic storm tracks, for example, in subtropical (Otkin and Martin, 2004; Evans and Guishard, 2009; Guishard et al., 2009; Evans and Braun, 2012) and in polar environments (Mansfield, 1974; Rasmussen, 2003; Zahn and von Storch, 2008; Simmonds and Rudeva, 2012). During their life cycle, the deepening of extratropical cyclones is supported by a combination of upper- and lower-level forcing mechanisms and the relative contributions of these forcing mechanisms depends on the cyclone environment. In general, strong deepening is often driven by upper-level vorticity advection and flow divergence ahead of a developing upper-level trough, which results in large-scale upward motion (Sutcliffe, 1947; Hoskins et al., 1978; Trenberth, 1978; Hoskins and Pedder, 1980; Deveson et al., 2002; Gray and Dacre, 2006). At lower levels, it is diabatic heating (Rogers and Bosart, 1986; Kuo et al., 1990; Reed et al., 1992; Davis, 1992; Whitaker and Davis, 1994; Stoelinga, 1996; Schemm et al., 2013; Binder et al., 2016) and high baroclinicity (Charney, 1947; Eady, 1949; Lindzen and Farrell, 1980; Rogers and Bosart, 1986) that accelerate the deepening of extratropical cyclones. Consequently, different categories of cyclones have been established based on different dominant forcing mechanisms (Petterssen and Smebye, 1971; Deveson et al., 2002; Hart, 2003; Gray and Dacre, 2006; Dacre and Gray, 2013; Graf et al., 2017; Catto, 2018). In practice, however, the separation between the different forcing mechanism is often not clear cut, and extratropical cyclones tend to grow within a wide range of these forcing mechanisms.

Further, the deepening of extratropical cyclones is inherently connected with the direction of propagation. It is longstanding knowledge that extratropical cyclones tend to propagate poleward (Hoskins and Hodges, 2002), but in contrast to tropical cyclones, the mechanisms driving their poleward motion have received enhanced attention only in recent years (Gilet et al., 2009; Rivière et al., 2012). An upper-level flow anomaly (corresponding to the upper-level trough or positive potential vorticity anomaly) that is located to the west of the surface cyclone mutually interacts, by means of an induced circulation, with the positive surface temperature anomaly (corresponding to a surface PV anomaly). This interaction enhances the poleward heat flux that in turn enhances the baroclinic growth (Hoskins et al., 1985; Coronel et al., 2015; Tamarin and Kaspi, 2016). The combined upper and lower-level circulations, which include the so-called $\beta$-drift, advects the cyclone center poleward (Gilet et al., 2009; Rivière et al., 2012; Tamarin and Kaspi, 2016). Rapid deepening and poleward propagation are therefore inherently connected.

In this climatological analysis, we quantify the regional variability of the upper- and lower-level forcing mechanisms during the growth phase, i.e., between genesis and maturity, of extratropical cyclones and systematically link this information with the direction of propagation. Specifically, the focus is set on: (i) the strength of the upper-level forcing, as measured by the quasi-geostrophic (QG) $\omega$-equation (Hoskins et al., 1978); (ii) the strength of the low-level baroclinicity, as measured by the Eady growth rate (EGR) (Lindzen and Farrell, 1980); and (iii) the link with the cyclone deepening rates at the surface and propagation directions obtained from a feature-based cyclone track climatology (Wernli and Schwierz, 2006; Sprenger et al., 2017). In this study, we focus on the mechanisms that reflect the dry-dynamic forcing of the cyclone deepening. Diabatic processes are only

indirectly accounted for via their influence on the low-level baroclinicity. The influence of diabatic processes has already been analysed in previous climatological studies (Čampa and Wernli, 2012; Boettcher and Wernli, 2013; Büler and Pfahl, 2017).

The main motivation for the selection of these two variables results from the classical picture of the extra-tropical cyclone development. The classical picture contains two key ingredients, which are a low-level zone of enhanced baroclinicity, defined as the meridional temperature gradient divided by the static stability, and an upper-level forcing of vertical lifting that triggers baroclinic growth (e.g. Petterssen and Smebye, 1971; Hoskins and Pedder, 1980; Browning et al., 1990; Semple, 2003). We have chosen to use the Eady growth rate and the QG $\omega$-equation because these two allow for a quantification of the strength

of these two ingredients. The QG $\omega$ quantifies the upper-level forcing of vertical motion, and hence the growth trigger, and the EGR is a measure for the baroclinic growth potential. Note, multiplication of the EGR with the eddy heat flux would yield a measure for the baroclinic conversion rate (Eq. 4 in Schemm and Rivière (2019)). We can thus expect strong growth in situations of high EGR and QG $\omega$ but we also expect a wide range of possible EGR and QG $\omega$ environments in which cyclones growth occurs. Our aim is to quantify this range, relate it to the observed cyclone growth rate and to identify regional

differences.

    The relevance of this research topic is highlighted by the fact that the environment and the different forcing factors, which drive extratropical cyclone deepening and the direction of propagation, are expected to change as the climate warms (Shaw et al., 2016; Catto et al., 2019). Indeed, climate projections suggest a decrease in low-level baroclinicity due to an amplified warming at higher latitudes, a process known as the Arctic Amplification. On the other hand, the increased water storage

capacity of the atmosphere in a warmer climate suggests a potential increase in the latent heat release and thus (positive) diabatic impact on cyclone development. Both changes seem to be engaged in a tug-of-war (Catto et al., 2019) resulting in a fuzzy picture of how extratropical cyclones will change in a warmer climate. Changes in the dry upper-level forcing are not well known. Overall, extratropical cyclones are projected to become slightly stronger and less frequent, though the number of extreme cyclones likely increases (Lambert and Fyfe, 2006; Ulbrich et al., 2008; Bengtsson et al., 2009; Zappa et al., 2013).

At the same time, the storm tracks are projected to shift poleward (Bengtsson et al., 2009; Chang and Guo, 2012), which could be a result of enhanced poleward propagation (Tamarin and Kaspi, 2016).

    Our study is also an attempt to create a baseline of the lower- and upper-level dry-dynamic forcing and its variability under present day climate conditions, which may prove useful in the assessment of its future changes. The study is structured as follows. In section 2, the used data sets (cyclone tracks and forcing factors) are introduced. In section 3, we address the cyclone

growth, its link to the forcing factors and quantify their regional variability. Section 4 considers in detail the direction of cyclone propagation, as it is linked to cyclone deepening and the forcing factors. Finally, the study concludes, in section 5, with a summary, some caveats and an outlook.

## 2 Data and Methods

### 2.1 ERA-Interim and dry-dynamic forcing

The analysis is based on the ERA-Interim data set from 1979 to 2016 (Dee et al., 2011), provided by the European center for Medium-Range Weather Forecasts (ECMWF). The meteorological fields are available on 60 hybrid-sigma model levels, have a spatial resolution of 80 km (T255 spectral resolution) and are temporally structured into 6-hourly time steps. We have interpolated the fields onto a global longitude-latitude grid with a resolution of $1° \times 1°$. The analysis in this study is restricted to the Northern Hemisphere north of $20°$ N and covers the extended winter (October–March).

The ERA-Interim data is used to identify cyclones based on sea level pressure (SLP). Additionally, secondary fields such as Eady Growth Rate (EGR) and $\omega$ forcing (QG$\omega$) are calculated according to Graf et al. (2017). We exclusively considered QG$\omega$ on 500 hPa and restricted it to the forcing from levels above 550 hPa, i.e., it represents upper-level forcing. EGR, on the other hand, is representative for low- to mid-tropospheric baroclinicity (850 to 500 hPa). More formally, QG$\omega$ is calculated by inverting the QG $\omega$-equation, which in the $\boldsymbol{Q}$-vector formulation reads as follows (Davies, 2015):

$$\left( \sigma \nabla^2 + f_0^2 \frac{\partial^2}{\partial p^2} \right) \omega = -2 \nabla \cdot \mathbf{Q}$$

Here, $f_0$ denotes the Coriolis parameter and $\sigma$ the static stability in pressure coordinates, which is defined by

$$\sigma(p) = -\frac{p}{R} \frac{T_v}{\theta} \frac{d\theta}{dp} .$$

The static stability is not constant in the domain. However, to avoid numerical stability problems in situations of near neutral or negative static stability, a 1d vertical profile of the static stability is used in the inversion of the $\boldsymbol{Q}$-vector equation. The profile
is computed as a simple horizontal domain average. It has been shown in previous case studies that this pragmatic choice has only a marginal influence on the final outcome of the layer averaged upper-level forcing (Graf et al., 2017). The $\boldsymbol{Q}$ vector is defined by

$$\mathbf{Q} = \begin{bmatrix} \frac{\partial \mathbf{V}_g}{\partial x} \cdot \nabla \left( \frac{\partial \phi}{\partial p} \right) \\ \frac{\partial \mathbf{V}_g}{\partial y} \cdot \nabla \left( \frac{\partial \phi}{\partial p} \right) \end{bmatrix}$$

where $\mathbf{V}_g$ denotes the geostrophic wind and $\phi$ the geopotential. The forcing from upper levels only is obtained by setting the
divergence $\nabla \cdot \boldsymbol{Q}$ to zero for pressure levels from the surface to 550 hPa. The numerical details of the iterative inversion of the the QG $\omega$-equation follows the description in Stone (1968); Pascal and Sprenger (2009).

The formal definition of EGR is (Lindzen and Farrell, 1980),

$$\text{EGR} = 0.31 \frac{f}{N} \frac{\partial |\mathbf{u}|}{\partial z} \text{ with } N = \left( \frac{g}{\theta} \frac{\partial \theta}{\partial z} \right)^{1/2} ,$$

where $|\mathbf{u}|$ is the magnitude of the horizontal wind speed, $\theta$ potential temperature and $z$ height. The discretized form used in
this study represents the layer between 850 and 500 hPa,

$$\text{EGR} = 0.31 \frac{f}{N_{500-850}} \left[ \left( \frac{u_{500} - u_{850}}{z_{500} - z_{850}} \right)^2 + \left( \frac{v_{500} - v_{850}}{z_{500} - z_{850}} \right)^2 \right]^{1/2} \tag{0}$$

where $u_X$, $v_X$, $z_X$ are the two horizontal wind components and the height at the pressure level $X$, respectively, and $N_{500-850}$ is a pressure-weighted average of the Brunt-Väisälä frequency, which is computed on ERA-Interim model levels and later vertically averaged between the two pressure levels. The two variables are not fully independent of each other, since the temperature gradient affects the EGR and the $\boldsymbol{Q}$. The omega forcing is stronger in the presence of high baroclinicity. Because we analyze low-level EGR it is not the same horizontal temperature gradient that intervenes in the $\boldsymbol{Q}$ vector, which is analyzed at upper levels.

The forcing factors will be determined along all cyclone tracks, i.e. the geographical position of minimum SLP (see section 2.2 below). It is, however, not reasonable to only consider the values of QG$\omega$ and EGR at the cyclone's center, which is defined by the location of the minimum SLP, because the cyclone growth and propagation is determined by its larger environment. Hence, we calculated the mean value of EGR and QG$\omega$ within a 1000 km radius around the cyclone center. For QG$\omega$, two mean values were calculated: one by only considering negative values (corresponding to forcing of upward motion), and a second one considering only positive values (forcing of downward motion). In this way, we take into account that QG$\omega$ often appears as a dipole, with the effect that the two dipole parts counterbalance each other if a simple mean is calculated.

## 2.2 Cyclone climatology and time normalization

To obtain the cyclone climatology, the identification and tracking algorithm by Sprenger et al. (2017) was employed, which is a slightly modified version of the algorithm introduced by Wernli and Schwierz (2006). First, the algorithm scans the grid points for SLP minima defined as having a lower value than the eight neighbouring SLP values on the grid. Secondly, the cyclone extent is determined by the outermost closed SLP contour encompassing the identified SLP minima (assuming a 0.5 hPa interval). To exclude spurious, small-scale SLP minima, the enclosing contour has to exceed a minimum length of 100 km, otherwise the SLP mimimum is discarded. In some cases the outermost SLP contour contains more than one SLP minimum. If the distance between two SLP minima within the same outermost enclosing SLP contour is less than 1000 km, they are attributed to the same cyclone cluster, creating a multi-center cyclone. In this case only the lowest SLP minimum is kept and the others are disregarded. The central SLP value and its geographical coordinates are stored, and subsequently used to determine cyclone tracks with cyclogenesis and lysis at the first and final time step of the track, respectively. As in Sprenger et al. (2017), the tracks have to exceed a minimum lifetime of 24 h from genesis to lysis.

During their life cycle, cyclones can undergo a process called 'cyclone splitting', which occurs when a cyclone (or a multi-center system) breaks up and forms two (or more) cyclones that then extend from the same origin as two separate cyclone tracks. As a consequence, the newly formed cyclone typically experiences only decay characterised by an increasing SLP over the course of its life cycle. To eliminate this subcategory of cyclones from the data, all cyclones that exhibit their minimum SLP value either at genesis or at genesis + 6 hours are disregarded. In this way, the analysis is restricted to cyclones with an archetypal pressure evolution, i.e., starting with higher pressure at genesis than attained during maturity.

Because the lifetime of cyclones can extend from 24 h (by the requested minimum duration; see above) to several days and it is therefore difficult to compare different cyclone life cycles, the method of Schemm et al. (2018) is applied to normalise the cyclone lifetimes. More specifically, three main time stamps are determined: $t_{genesis}$ (time of genesis), $t_{max}$ (time of maximum

intensity, i.e., minimum SLP), and $t_{lysis}$ (time of lysis). Then $\Delta t_1$ is computed as the difference between $t_{genesis}$ and $t_{max}$, which results in a negative value:

$$\Delta t_1 = t_{genesis} - t_{max} \tag{0}$$

Next, the difference between $t_{lysis}$ and $t_{max}$ (indicated as $\Delta t_2$) is determined, which results in a positive value:

$$\Delta t_2 = t_{lysis} - t_{max} \tag{0}$$

For a certain time $t$ between $t_{genesis}$ and $t_{max}$ we can calculate a normalised time $t_{norm}$:

$$t_{norm} = \frac{t_{max} - t}{\Delta t_1} \tag{0}$$

The same is done for time $t$ between $t_{max}$ and $t_{lysis}$:

$$t_{norm} = \frac{t - t_{max}}{\Delta t_2} \tag{0}$$

Hence, in this normalized time frame, $t_{norm} = -1$ corresponds to cyclogenesis, $t_{norm} = 0$ to the time instance of minimum SLP, and $t_{norm} = +1$ to cyclolysis. In the remainder of the study, $t$ always refers to this normalized time. All of the analysis in section 3 and 4 will be restricted to the phase with normalized times between -1 and 0, i.e., the focus is on the cyclone's life cycle between genesis and the time instance of deepest SLP, i.e., the cyclone growth phase.

Finally, three geographical regions in the Northern Hemisphere are particularly considered in this study: North Pacific (125°-180°E and 25°-65°N); North Atlantic (75°W-0°and 25°-65°N); North America (120°-75°W and 25°-65°N). To be attributed to one of these three target regions, a cyclone must be located inside the respective latitude-longitude box at the time of genesis.

## 2.3 Dry-dynamic forcing categories

In this study, we frequently employ a 2D histogram, which has EGR on the x-axis and QG$\omega$ on the y-axis. The histogram consists of 49 linearly distributed bins defined by a range of EGR and QG$\omega$ values such that each bin is characterized by a specified EGR and QG$\omega$ forcing. Each six-hour time interval during a cyclone growth phase is classified according to its EGR and QG$\omega$ values such that each bin is populated by a multitude of cyclone time segments from various cyclone tracks. The color shading in each of the bins represents the mean value over all time steps of a specific cyclone characteristic, for example, the deepening rate. Figure 1 shows such a histogram where the mean represents the average per bin of all 12-hour changes in mean sea-level pressure ($\Delta$SLP). Consideration is given to all $\Delta$SLP changes during the growth period of every life cycle and not only to the maximum change. Therefore, every cyclone contributes with several time steps to the statistics and positive $\Delta$SLP values are accepted as long as they occur prior to the time of maximum intensity.Positive values indicate that the deepening between genesis and maximum intensity is not linear. If, for instance, a 12-hour $\Delta$SLP value of 6 hPa is associated with an EGR of 1.2 day$^{-1}$ and a QG$\omega$ value of -0.01 Pa s$^{-1}$, it contributes to the lower right corner of the histogram. The 12-hour $\Delta$SLP values in every bin will vary between different cyclones and each bin is therefore populated by a distribution of 12-hour $\Delta$SLP values. This is illustrated in Figure 1 for the four corners of the histograms. If not otherwise stated, the color shading

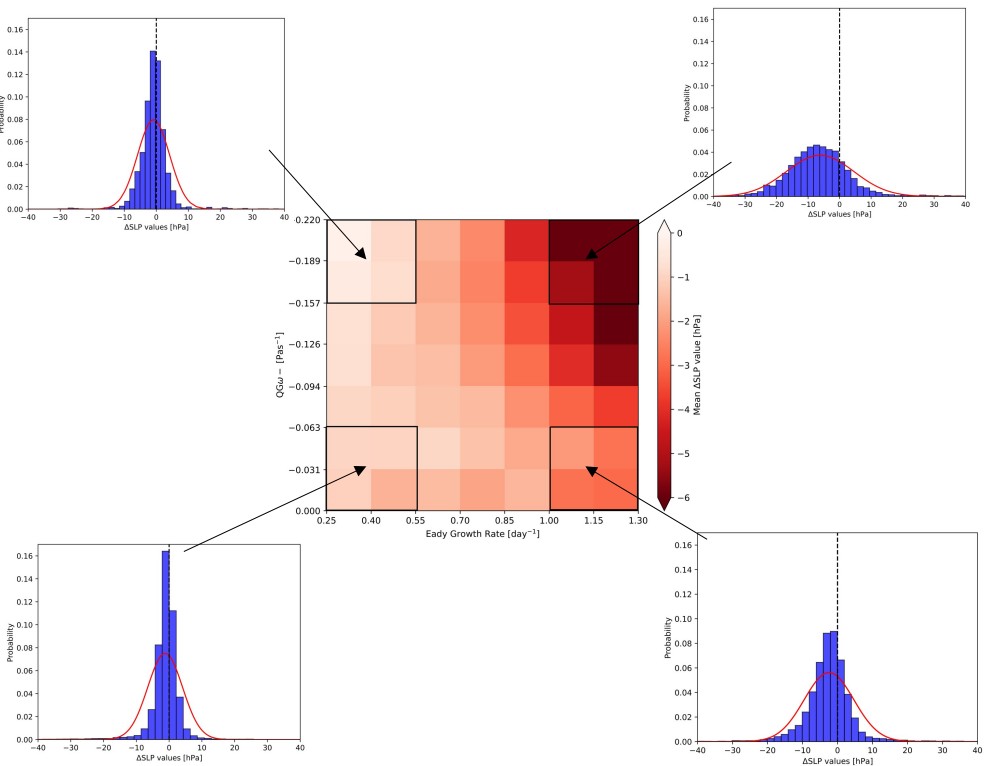

**Figure 1.** Exemplary 2D forcing histogram with EGR (in day$^{-1}$) on the x-axis and QG$\omega$ (in Pas$^{-1}$) on the y-axis. Colours show the mean $\Delta$SLP value for each bin (darker (lighter) red highlights bins containing time intervals with stronger (weaker) mean deepening rates ($\Delta$SLP)). For every corner of the histogram (framed in black, consisting of 2x2 bins), the distribution of $\Delta$SLP values is shown. The four boxes are used to define four forcing categories: Q↓E↓, Q↓E↑, Q↑E↓ and Q↑E↑ (see text for details).

represents the mean values. The distributions of the 12-hour $\Delta$SLP values resemble Gaussian distributions, but still differ in their shapes: some are narrow and centered around 0 hPa, whereas others are more broadly distributed. However, none of the distributions in Fig. 1 (and also in the results of section 3 and 4) is strongly skewed with long tails to one end. The number of cyclone track segments in each bin is of the order of several thousands, except for the bins close to the corners of the 2D

histogram. The specific numbers of cyclone track segments and therefore the number of $\Delta$SLP values can be seen in Fig. S1 in the supplement.

    The lower-left corner of the histogram represents low QG$\omega$ and low EGR forcing. Conversely, high QG$\omega$ and high EGR forcing is located in the upper-right corner. The lower-right (upper-left) corner represents cyclone growth in environments characterized by low (high) QG$\omega$ and high (low) EGR forcing. The four corners of the histogram are used to define four

forcing categories, which is discussed in more detail in section 3. More specifically, the box in the lower left corner, for

example, is referred to as Q↓E↓ (cyclone growths in a low QGω Q↓ and low EGR E↓ environment) and the other boxes are labelled accordingly.

## 3 Dry-dynamic forcing during cyclone growth

In this section, we study the regional variability of the forcing mechanisms during the growth phase of cyclones. We start with
the geographical distribution of the four categories Q↓E↓, Q↓E↑, Q↑E↓ and Q↑E↑ that were introduced in section 2.3, then proceed in section 3.2 with a detailed analysis of the forcing histogram and finally, in section 3.3, we discuss cyclone-centered composites of QGω and EGR.

### 3.1 Geographical distribution of dry-dynamic forcing

In this section, consideration is given to the geographical distribution of the four forcing categories. Density plots are created
by considering all time steps during the cyclones' growth phase and by computing their inclusiveness to one of the four forcing categories (Q↓E↓, Q↓E↑, Q↑E↓ or Q↑E↑) and the corresponding latitude-longitude location. The outcome is a remarkably distinct geographical distribution of the occurrence of the four forcing categories (Fig. 2). The four category-specific plots show a probability density distribution, which integrates to 1. Each forcing category has unique hot spots, which are discussed in the following.

We start our discussion of the forcing-based categories (Fig. 1) with category Q↑E↓ (Fig. 2a), which is mostly confined within a latitudinal band between 25° and 40° N. The major hot spot is located over the North Atlantic off the west coast of Northern Africa and spanning over the Mediterranean. A second but less dense region is discernible in the Pacific off the U.S. west coast, reminiscent of kona lows (Simpson, 1952; Morrison and Businger, 2001; Moore et al., 2008). The downstream regions are eventually related to secondary or downstream cyclogenesis, though one would expect high EGR due to the trailing
cold fronts typically involved in secondary cyclogenesis (Schemm and Sprenger, 2015; Schemm et al., 2018; Priestley et al., 2020). Over the Atlantic, this category comprises also the deepening of subtropical cyclones, which form under strong QGω forcing, provided by equatorward pushing intrusions of high-PV air (Caruso and Businger, 2006) and a weak baroclinic zone.

    The next category Q↑E↑ (Fig. 2b) has two distinct hot spots: one northeastward orientated band reaching from North America to Norway with the maximum frequency northeast of Nova Scotia in the North Atlantic, and the other hot spot off the coast
of Japan. Both are located slightly poleward of the identified hot spot in Q↓E↑ (Fig. 2d). We hypothesize that early during the life cycle time steps are categorized into Q↓E↑ (Fig. 2d), while afterward during the main deepening period both forcings contribute to the deepening and the corresponding time steps are categorized into Q↑E↑ (Fig. 2b). We will come back to this hypothesis in the next section.

    For category Q↓E↑ (Fig. 2d), the regions where the forcing occurs most frequently are over North America and partly over
the western North Atlantic, and further the southern tip of Greenland and parts of central Asia. Another prominent hot spot is located over the Pacific ocean off the coast of Japan. Over North America, the maximum is located near 50°N and therefore north of the cyclogenesis region downstream of the southern Rocky Mountains in the U.S. (see Fig. 5c in Hoskins and Hodges,

2002). It is connected to cyclone deepening in the lee of the Canadian Rocky Mountains and the formation of 'Alberta clipper' cyclones (Chung and Reinelt, 1973; Thomas and Martin, 2007). The southern tip of Greenland is a well-known cyclogenesis hot spot (Hoskins and Hodges, 2002; Wernli and Schwierz, 2006) and high baroclinicity in this region is connected to the steep slopes of the Greenland shelf. Off the coast of Japan, high baroclinicity is maintained by the Kuroshio sea-surface temperature front and the density maximum is located slightly equatorward of the maximum of the category Q↑E↑ (Fig. 2b).

Finally, for category Q↓E↓ (Fig. 2c) the highest density is located over the subtropical Atlantic spanning in a horizontal band from the Gulf of Mexico to North Africa, covering parts of the Mediterranean Sea and extending downstream into the Middle East, with a local maximum over Iran, and even further downstream over China and into the East China Sea. Over Asia, there is an additional hot spot region upstream of Kamchatka. These two branches over Asia correspond well with the two seeding branches of the North Pacific storm track described by Chang (2005). The cyclones along the southern branch, in contrast to the northern one, are known to be diabatically driven in the early life cycle stage (Chang, 2005). The Hudson Bay in Canada is another localized region where cyclone growth occurs in a low QG$\omega$ and EGR forcing environment. Surface cyclone deepening in this region is often connected to the development of a tropopause polar vortex also known as upper-level cut-off low (Gachon et al., 2003; Cavallo and Hakim, 2009; Portmann et al., 2020). The deepening maximum over the subtropical North Atlantic comprises subtropical cyclone development (Caruso and Businger, 2006) and because it is located north of a known Hurricane genesis region, it might also contain some recurring tropical cyclones (Landsea, 1993; McTaggart-Cowan et al., 2008). The smaller maximum over western North Africa indicates the deepening of African easterly waves (Burpee, 1972), which often precede Hurricane formation over the subtropical North Atlantic (Landsea, 1993; Avila et al., 2000). It is noteworthy at this stage, that EGR and QG$\omega$ forcing is low relative to all other locations in the Northern Hemisphere where cyclone growth occurs, however, the observed EGR and QG$\omega$ forcing might be high relative to a local climatology (see supplement Fig. S2a and b). Furthermore, it is interesting to relate the the density distribution of the four categories (Fig. 2a-d) into context of the climatological relationship between EGR and QG$\omega$. For example, category Q↑E↑ (Fig. 2b) is found at the beginning of the North Atlantic and Pacific storm tracks. These are regions where the correlation between QG$\omega$ and EGR (see supplement Fig. S2c) is negative and also somewhat enhanced compared to mid- and east-oceanic regions, i.e., the correlation matches the expectation. However, the link to the other categories is not particularly strong. With respect to the correlation between EGR and QG$\omega$ (Fig. S2c) regardless of the four forcing categories, the correlation remains rather weak in the main North Atlantic and Pacific storm tracks, and the correlation is larger in the western part of the storm tracks, while it becomes smaller towards the east. Furthermore, the largest anti-correlation are found in the subtropics east of China and west of North America and Africa. Positive correlations are essentially restricted to the region east of the Himalayas.

Figure S3 in the supplement shows the winter climatology for QG$\omega$, as in Fig. S2b, but for negative (Fig. S3a) and positive (Fig. S3b) QG$\omega$ values separately. The patterns in both Figures are similar, which underlines the fact that the positive and negative anomalies often co-occur in QG$\omega$-dipoles. Still, there are noteworthy differences. One specific example is the positive pole in the eastern Mediterranean, which is located further to the west compared to its negative counterpart. This indicates that the positive and negative anomalies are most likely part of common weather systems, e.g., to the west and east of a short-wave trough.

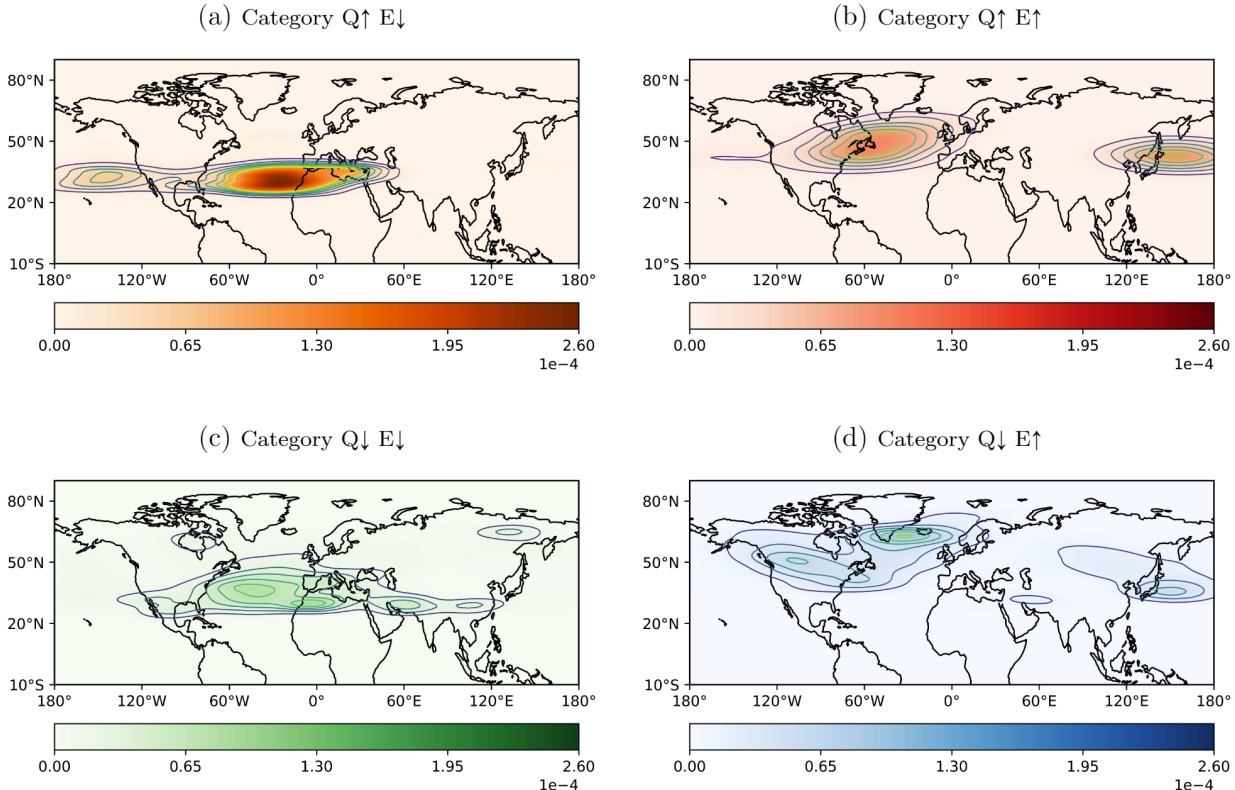

**Figure 2.** Normalized geographical density distribution (coloured contours and filled contours) of the four selected forcing categories: (a) high QGω forcing and low EGR (Q↑E↓), (b) high QGω and high EGR (Q↑E↑), (c) low QGω and low EGR (Q↓E↓), and (d) low QGω and high EGR (Q↓E↑). For the exact computation, see text.

In addition to the forcing distribution and climatology of EGR and QGω, category distributions as in Fig. 2 but for EGR and QGω separately could be considered. This is presented in Fig. S4 in the supplement, whereas these distributions are, for the most part, a combination of the categories shown in Fig. 2. For instance, Fig. S4a shows the geographical distribution for E↓, which is a combination of Q↑E↓ (Fig. 2a) and Q↓E↓ (Fig. 2c). However, for Q↑ (Fig. S4d), the major hot spot over the Atlantic off the coast of Northwestern Africa (seen in Fig. 2a) disappears due to a low density relative to other regions.

## 3.2 Temporal evolution of dry-dynamic forcing

In the previous section, we discussed the geographical distribution of the two dry-dynamic forcing mechanisms during the cyclone growth phase. In this section, we discuss their temporal evolution because the dominating forcing mechanism might change during the growth phase. Figure 3a shows the forcing histogram, as in Fig. 1, exclusively for the cyclones' growth phase and Fig. 3b shows the corresponding normalized life-cycle time $t_{norm}$ ($t_{norm} = -1$ corresponds to genesis and $t_{norm} = 0$ to

maximum intensity). Strongest deepening is depicted by dark red shading in the upper-right corner in Fig. 3a. It occurs, not too surprising, in a high QGω and high EGR environment. The deepening rates in a low QGω and low EGR environment (lower-left corner in Fig. 3a) are consequently low. However, the upper-left and lower-right corner of the forcing histogram differ: the deepening rates are larger in a high EGR and low QGω environment (lower right) compared to a low EGR and high QGω environment (upper left). Potentially this asymmetry is due to the connection between high EGR environments, for example along a surface front, and diabatic forcing as is the case for diabatic Rossby wave development (Boettcher and Wernli, 2013), however diabatic processes are intentionally disregarded in our analysis. The asymmetry between the two opposite corners could also point to a (potentiallly) subtle difference in EGR and QGω forcing. It indicates that EGR has a stronger influence on the deepening rates than QGω, and that baroclinic instability might be released as long as there is a reasonable (even moderate) amount of upper-level forcing. In short, moderate upper-level forcing (QGω) might be sufficient to trigger substantial deepening rates if EGR is high. In contrast, if EGR is low, weaker deepening rates result even if substantial upper-level forcing is discernible. Furthermore, Fig. S6 in the supplement shows the forcing histogram with the standard deviation of the $\Delta$SLP for each bin with color shading and the exact value in every bin. The Figure indicates that there is not a linear increase in the deepening rates with linearly increasing EGR and QG Omega. Because the standard deviation in each bin is larger than the difference between two consecutive mean values between two bins, we expect that very high values occur in a bin with lower mean value than that in is neighbouring bin. Bins in the lower-left corner (low EGR, low QGω) show standard deviation values between 4 and 6 hPa (Fig. S6), while the difference in mean values between the bins is less than 1 hPa (Fig. 3a).

The $t_{norm}$ histogram indicates that QGω forcing increases as a cyclone approaches its mature stage (white shading and $t_{norm} = 0$ in Fig. 3b), i.e., a period during which the upper-level trough intensifies. The most negative normalized times (i.e., closest to cyclogenesis) are found in the lower-right corner with high EGR, but low QGω forcing. Hence, it seems – and is intuitively reasonable – that on average EGR is high at genesis and early during the growth phase while high QGω forcing builds up until reaching maturity. In summary, we conclude that the four forcing categories differ not only in their geographical distribution, they also preferentially occur during different periods of the cyclone growth phase. More specifically, the temporal occurrence according to Fig. 3 is: Q↓E↑ occurs closest to genesis and Q↑E↓ closest to maturity. Hence, a cyclone progresses forward in time from genesis to maturity as EGR decreases and QGω increases.

Figure 4 further illustrates the mean forcing evolution with regards to the normalised time as previously presented in Figure 3b. The EGR (left y-axis in purple) as well as QGω (right y-axis in blue) are plotted against the normalised time, whereas $t = -1$ represents cyclogenesis and $t = 0$ the point of maximum intensity. The shaded area shows the range of 1 standard deviation. The mean evolution shown in Figure 4 is not representative of any one individual cyclone lifecycle. The figure only shows, for instance, that at the time of genesis (at time -1) cyclones are associated with intermediate EGR values and high QG omega values. This seems physically plausible because a low-level baroclinic zone (as expressed with EGR) is only one ingredient for cyclogenesis. The upper-level forcing (QG omega) might act as a trigger to release the baroclinic instability, and hence allow for the further cyclone deepening. Interestingly, immediately after genesis, e.g. at normalised time -0.8, the EGR values become larger and the QG omega values slightly weaker. This agrees with our aforementioned perception that the

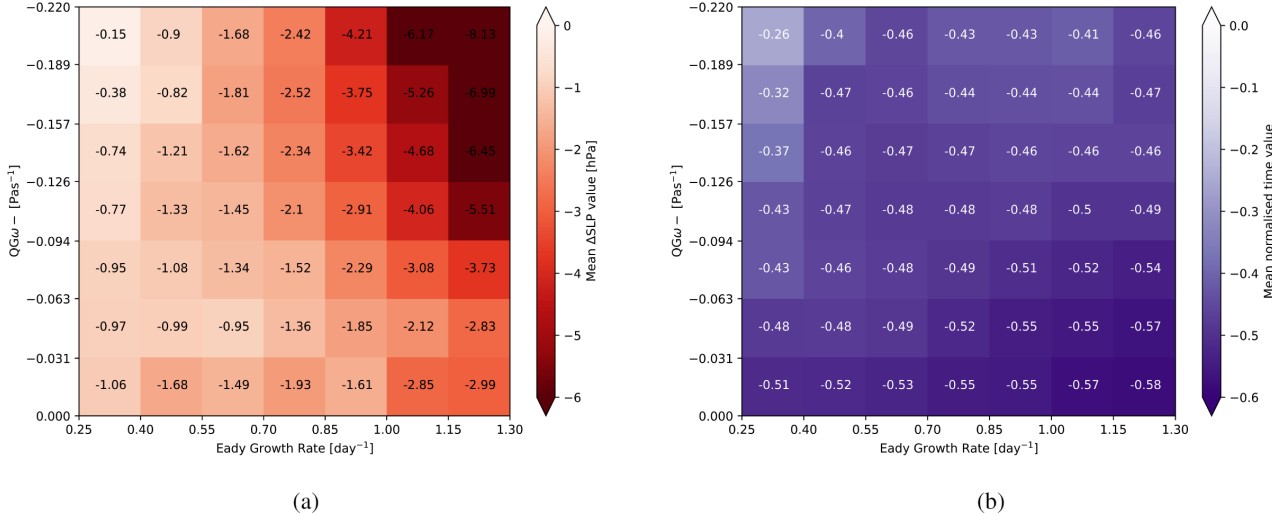

**Figure 3.** 2D forcing histograms with EGR (in day$^{-1}$) on the x-axis and QG$\omega$ (in Pa s$^{-1}$) on the y-axis. In panel (a), the mean of the 12-hour $\Delta$SLP distribution within each 2D bin is colored, with darker red colors indicating a stronger cyclone growth. In panel (b), the mean of the normalised time values is shown, with lighter purple colours for mean time values near the time instance of deepest SLP. The numbers in the bins give the exact values, corresponding to the color shading. For further details on the 2D histograms and time normalization, see section 2.

cyclone deepening is governed in the early phase of the growth period by the high EGR values. Only later, towards the phase of deepest SLP and when the cyclone has attained a mature state, the upper-level forcing becomes large again. The steady increase in QG omega between normalised times -0.8 and -0.2 thereby reflects the co-evolution of the near surface and the upper-level flow. Finally, near normalised time 0, both forcing factors steeply decrease, which, of course, makes sense since the cyclone has already reached its mature stage and starts to decay for times larger than 0. Figure S5 in the supplement relates the normalised time to the "real" mean time in hours before maximum intensity in order to give some context to the dimensionless normalised time. Moreover, Figure 4 shows that EGR values are dispersed only within a rather narrow range of values ( 0.67-0.9 day$^{-1}$), while QG$\omega$ values assume a much broader range from -0.04 to -0.16 Pas$^{-1}$ during the life cycle.

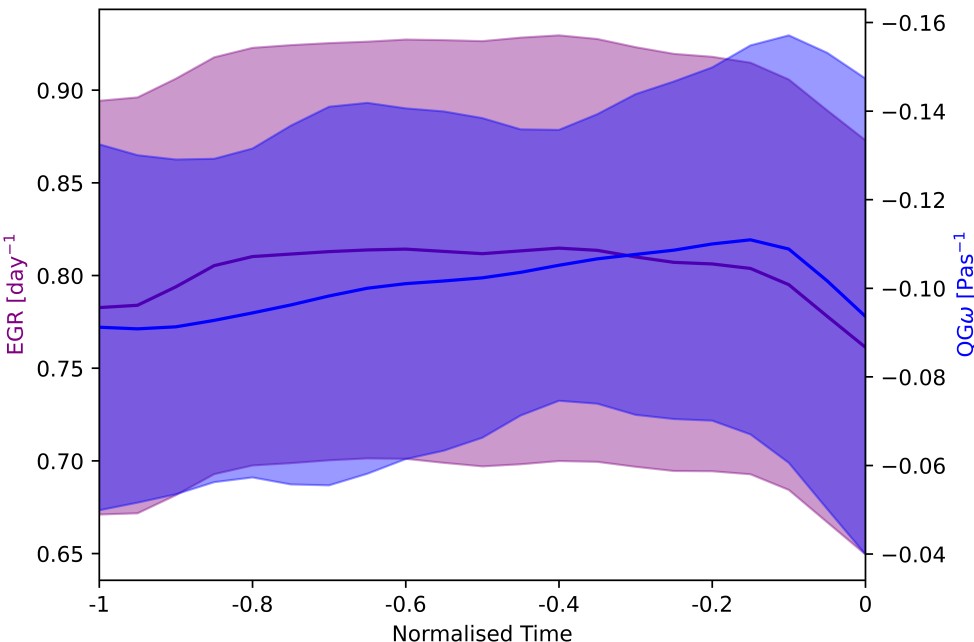

**Figure 4.** Mean evolution of EGR (purple, in day$^{-1}$) and QG$\omega$ (blue, in Pas$^{-1}$) over normalised time (dimensionless) during the intensifying stage. The shaded are represents the range of 1 standard deviation (0.5 STD on either side) for EGR (purple) and QG$\omega$ (blue).

### 3.3 Cyclone-centered composites

In this section, we turn our attention to the immediate surroundings of cyclones during their growth phase. This is done individually for each of the four forcing categories and for PV at 320 K, QG$\omega$ forcing and EGR (Figs. 5 and 6).

Category Q↑E↓ (Fig. 5a) displays a strong upper-level PV signal of up to 2.5 pvu resembling the structure of an upper-level PV cutoff. The upper-level PV maximum is essentially located exactly above the surface cyclone center, i.e., the whole flow situation is nearly barotropic and hence indicates that only weak further cyclone deepening can be expected. This is in agreement with the deepening rates in Fig. 3a and it also fits well to the normalized times in Fig. 3b, which now are outside the time window where the strongest deepening rates are expected ($t_{norm} > -0.5$). In accordance with the structure of the upper-level PV, QG$\omega$ exhibits a rather symmetric dipole, with descent to the west, ascent to the east and the surface cyclone

center slightly shifted towards the ascending pole (Fig. 5a). This reflects how sensitive the forcing at the cyclone center reacts to slight horizontal displacements. In this category, for example, the forcing of vertical motion occurs too far to the east of the cyclone center, resulting in weaker cyclone deepening than in category Q↑E↑ (Fig. 5b). Finally, the EGR signal in this category at the cyclone position (Fig. 6a) is clearly weaker than for the E↑ categories. Enhanced values are found to the southwest of the upper-level PV structure indicative of a local velocity maximum (a jet streak), but at the cyclone center EGR remains rather

small.

Category Q↑E↑ category depicts the strongest upper-level PV signal (Fig. 5b) among all four categories, with amplitudes reaching up to 3.5 pvu and a rather pronounced horizontal westward tilt of the upper-level PV maximum relative to the surface cyclone center. It resembles a well-developed trough located upstream of the surface cyclone center and it clearly fits well into the conceptual model of a deepening cyclone in a PV framework (Hoskins et al., 1985). In accordance with the strong upper-level PV signal, a strong QG$\omega$ dipole is discernible (Fig 5b): with the surface cyclone located slightly to the south of the QG$\omega$ maximum. The EGR maximum, on the other hand, is found to the southwest of the cyclone center (Fig. 6b) where we expect the trailing surface cold front. Given the strong forcing and the archetypal flow situation that is well known for many developing cyclones, it is no surprise that this category is characterized by the largest deepening rates (as seen in Fig. 3a).

Category Q↓E↓ displays a minor upper-level PV structure (Fig. 5c), resembling a PV cutoff centered above the surface cyclone's center attaining only a small amplitude of 1.5 pvu. The barotropic structure and small amplitude of the upper-level PV points to small deepening rates, in particular together with a weak QG$\omega$ forcing (Fig. 6c) and a uniform low EGR environment (Fig. 6c). Indeed, category Q↓E↓ exhibits the weakest deepening rates (Fig. 3a).

A completely different upper-level PV structure is discernible for category Q↓E↑ (Fig. 5d). The cyclone center is located on the flank of a southwest to northeast oriented band of enhanced PV gradients. The cyclone is likely located near the exit of a jet streak that forms upstream around the trough. This is in agreement with the existence of upper-level QG$\omega$ forcing (Fig. 5d) that is larger compared with Q↓E↓ (Fig. 5c) but lower compared with Q↑E↓ (Fig. 5b). In this meteorological scenario we expect enhanced EGR forcing, remembering that an upper-level jet by thermal wind balance must be associated with a significant horizontal temperature gradient beneath its core and hence also with a corresponding EGR signal by definition (see section 2.1). Indeed, this can be seen in Fig. 6d. The normalized times associated with this category (lower-right in Fig. 3b) indicate that the cyclone development is rather in an early stage, as one would expect from the upper-level PV structure that displays only a weakly developed trough and ridge.

## 4 Dry-dynamic forcing, deepening rates and propagation direction

While the previous section solely focused on the deepening rates of extratropical cyclones, consideration is now given to the connection between the dry-dynamic forcing, the deepening rates and the direction of propagation of the cyclone during its growth phase. The propagation direction at a time instance along a track is determined by taking the cyclone's six-hour displacement vector and determining the angle between this vector and a zonal vector, i.e., an angle 0 corresponds to eastward propagation and 90° to northward propagation.

### 4.1 Propagation angle and deepening rates

Figure 7 shows four different windroses for varying deepening rate regimes by means of the 12-hourly SLP changes. For example, Fig. 7a consists of propagation angles corresponding to time steps with deepening rates of $10\,\mathrm{hPa}\,12\,\mathrm{h}^{-1}$ or more. The rings indicate the frequency of a specific angle range (i.e., the windrose petal). For instance, Fig. 7a includes several petals, the longest of which points in northeastern direction or 45°. The corresponding petal reaches the outer ring of the plot

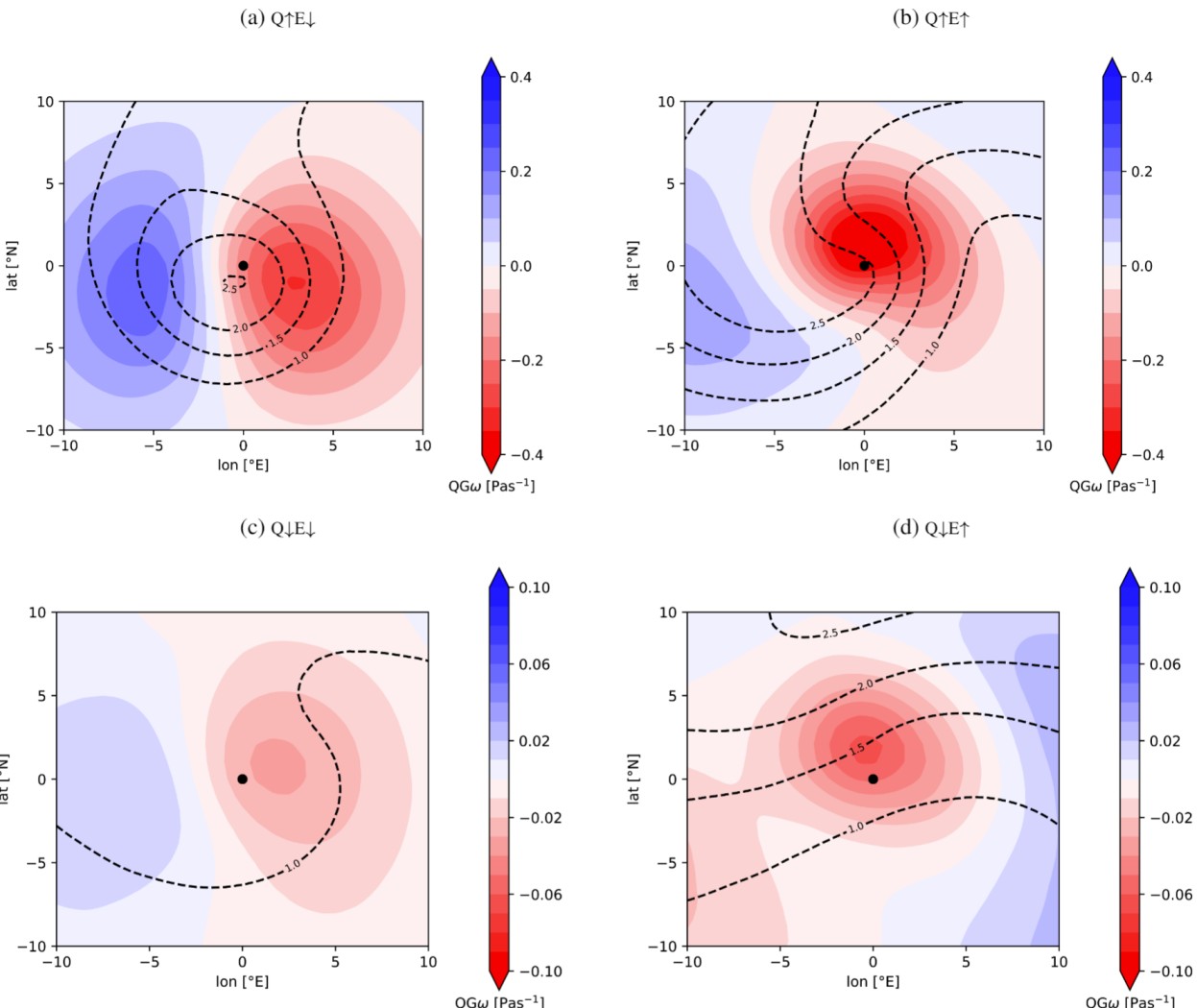

**Figure 5.** Cyclone-centered composites of upper-level QG$\omega$ (in Pa s$^{-1}$) during the growth phase ($-1 < t_{norm} < 0$) for the four forcing categories: (a) Q↑E↓, (b) Q↑E↑, (c) Q↓E↓ and (d) Q↓E↑; the black dot represents the cyclone center (SLP minimum). Additionally, contour lines of upper-level PV (in pvu) on the 320 K isentrope are shown.

indicating that 34.2% of all cyclones that deepen at a rate of 10 hPa 12 h$^{-1}$ or less propagate into the northeastern direction. Therefore, the number and size of the windrose petals indicate the relationship between the cyclone deepening and the direction

of propagation. The results for weaker deepening (-10 hPa $<$ $\Delta$SLP $<$ -6.5 hPa) (Fig. 7b) indicate a dominant northeast-oriented propagation direction although the east-northeast petal is longer than in 7a. For even weaker deepening (-6.5 hPa $<$ $\Delta$SLP $<$ -3 hPa)(Fig. 7c), most of the values are also within the northeastern direction, however, the largest petal is found in the east-northeast section (Figure 7c). Moreover, the petal in eastern direction, for these weaker deepening rates, is significantly larger

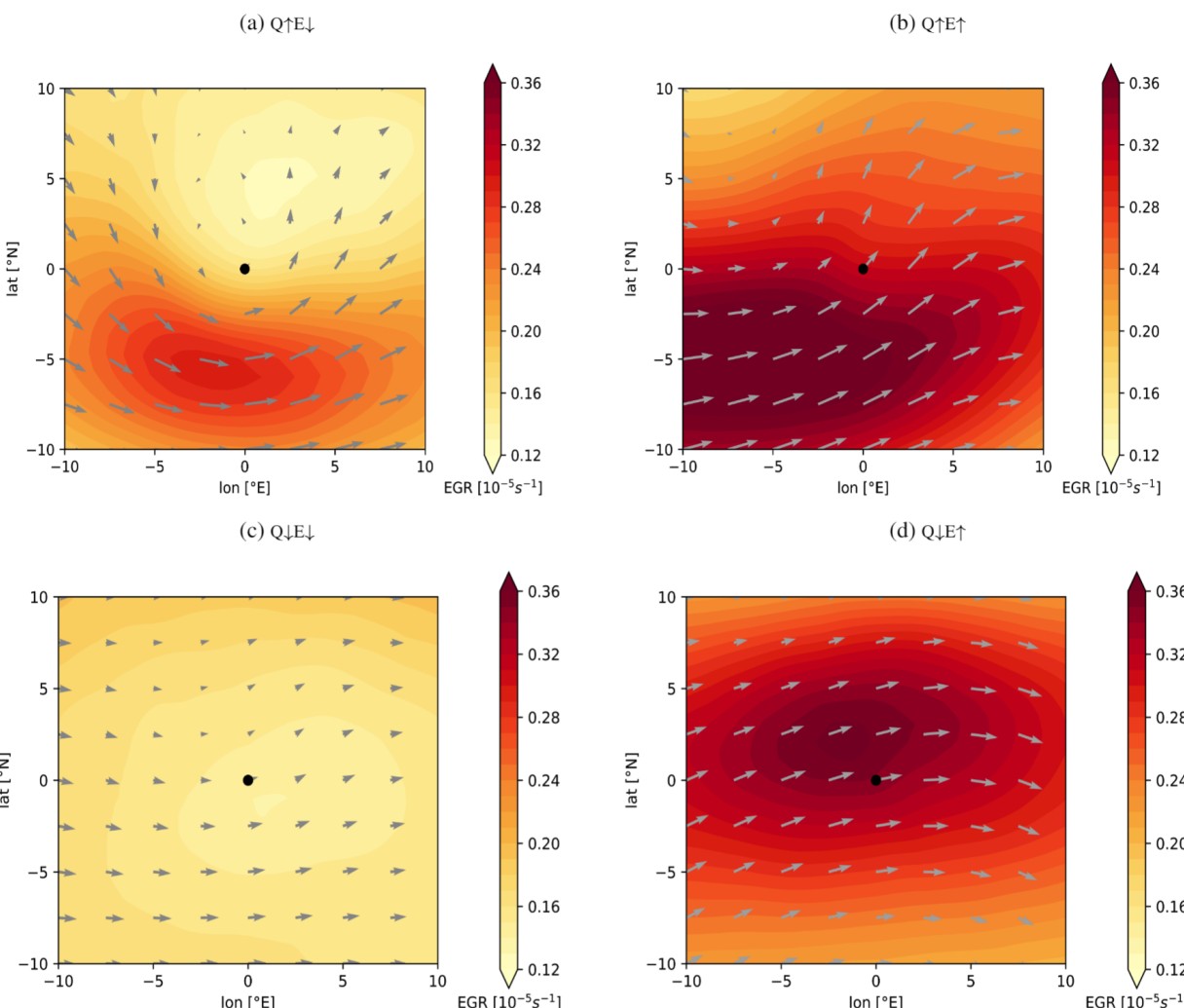

**Figure 6.** As in Figure 4, but for EGR (in $10^{-5}\,s^{-1}$). Grey arrows represent the windfield at 300hPa.

than in Figs. 7a and 7b. Finally, a shift toward zonal (eastward) propagation angles is found for the weakest deepening rates (-3 hPa $< \Delta$SLP $< 0$ hPa)(Fig. 7d), where the two petals in east and east-northeastern direction are the most prominent ones, each representing approximately 20% of the angle values. Overall, we can summarize that while during their growth phase cyclones go through different magnitudes of deepening rates, during the times a cyclone experiences higher deepening rates it tends to propagate more poleward.

Next, consideration is given to the geographical distribution of the direction of propagation. In Fig. 8 blue colours indicate a poleward propagation and red colour represents equatorward propagation. Areas with climatological equatorward propagation are sparse and spatially confined to regions downstream of mountain ranges, e.g., downstream of the Tibetan Plateau, leeward

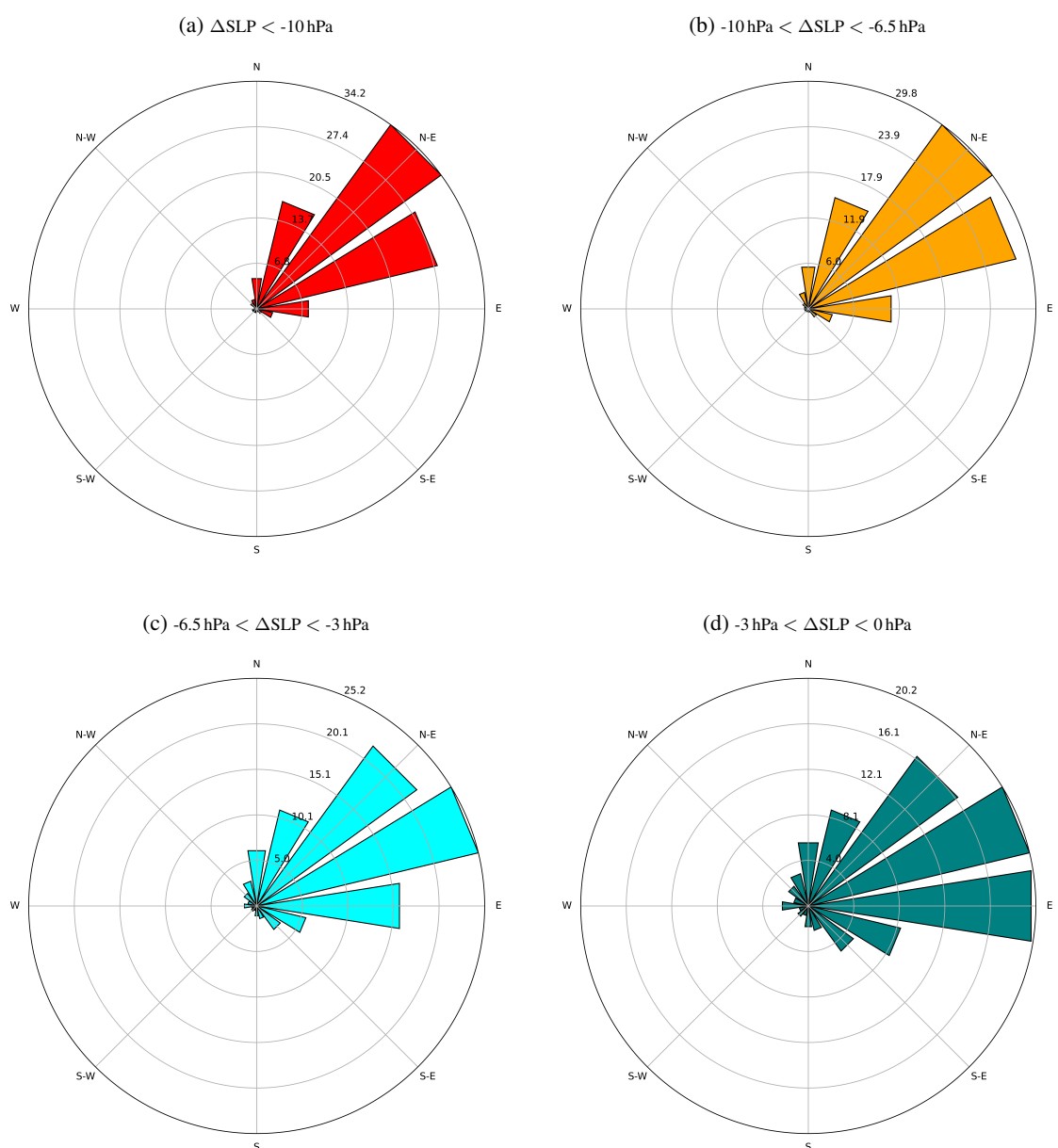

**Figure 7.** Windrose plots showing the frequency (in percentage) of angle values for all 6-hour track sections for different degrees of 12-hour cyclone deepening rates: (a) ΔSLP values below -10 hPa, (a) ΔSLP values between -10 and -6.5 hPa, (c) ΔSLP values between -6.5 and -3 hPa, and (d) ΔSLP values between -3 and 0 hPa.

of the Himalayas and over North America in the lee of the Rocky Mountains. Over Europe, equatorward propagation can be

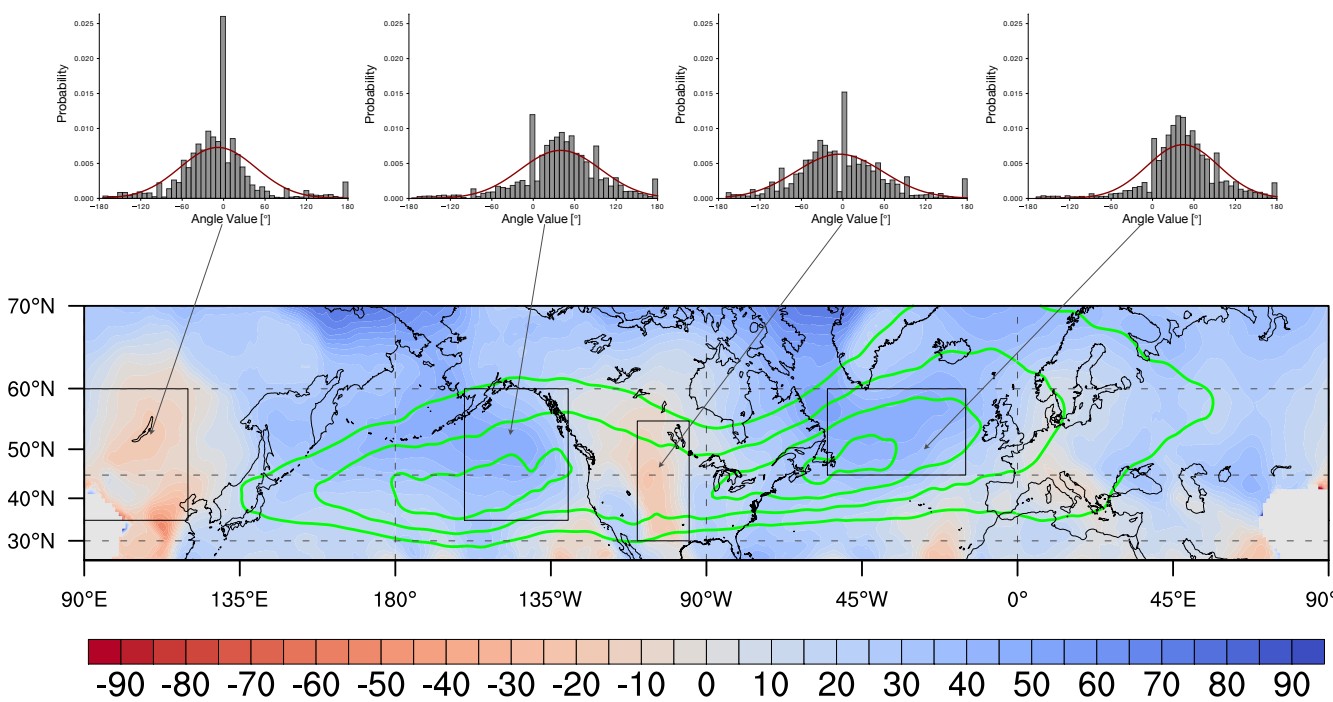

**Figure 8.** Mean propagation angles (in degrees) in extended winter (October-March), averaged over a 500 km radius. Positive values (blue) correspond to poleward propagation, negative ones (red) to equatorward propagation. Additionally, four regions with distinct propagation signals are outlined by black boxes and the distribution of propagation angles for all cyclones passing the region are shown, complemented by a Gaussian fit as a red line. The green contours represent eddy kinetic energy (10-day lowpass filter) at the 500 hPa level and provide the extent and intensity of the storm tracks.

found leeward of the Alps. Hence, mountain ranges are able to favor equatorward propagation in their lee (in agreement with the formation of stationary lee troughs).

A tendency for poleward propagation is characteristic for the North Atlantic and North Pacific storm tracks. It is striking, and in agreement with earlier studies (Gilet et al., 2009; Rivière et al., 2012; Tamarin and Kaspi, 2016), how the region of maximum deepening in the two oceanic basins coincide with local maxima in mean poleward propagation angles. Downstream of these maxima, in particular for the North Atlantic storm track, the poleward tendencies steadily decrease and over Europe the tendencies attain rather zonal values. Besides the storm track regions, additional distinct regions exhibit positive mean

propagation angles: for instance, over California to the west of the Rocky Mountains, to the east of Greenland, over the Black Sea, to the east of Lake Baikal, over North East Siberia and the nearby the Arctic Sea. It remains to be studied in a refined analysis to which degree these tendencies are determined by orographic effects or other forcings.

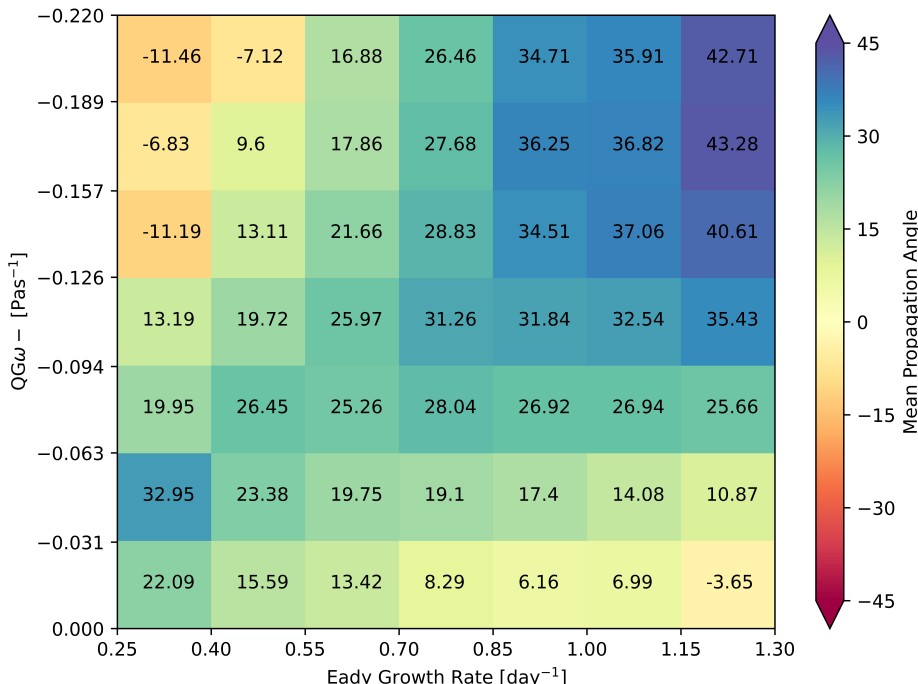

**Figure 9.** 2D forcing histogram as in Fig. 3a but depicting the mean propagation angle in color (in degrees; positive toward the poles, negative toward the equator) against QG$\omega$ forcing (Pa s$^{-1}$) and Eady Growth Rate (day$^{-1}$).

Of course, as seen in the histograms of propagation angles for the outlined four regions in Fig. 7, each region is characterized by a rather wide spread of possible propagation angles. For instance, in the North Pacific the mean propagation angle peaks near 45°, but smaller and higher values often occur. Even higher northward propagation angles, in the mean, are found in the box over the western and central North Atlantic. This is in agreement with the fact that the North Atlantic storm track is more tilted towards the northeast with increasing longitude compared to the North Pacific storm track (Hoskins and Hodges, 2002; Wernli and Schwierz, 2006).

### 4.2 2D forcing diagram for cyclone propagation

To determine the relationship between the dry-dynamic forcing (QG$\omega$ and EGR) and the direction of propagation a histogram similar to the one in Fig. 3a is computed (Fig. 9). The first noticeable characteristic of the histogram is the fact that the upper-right corner displays the strongest poleward propagation direction under strong dry-dynamic forcing. This was also the corner with the highest 12-hour $\Delta$SLP deepening rates (Fig. 3a) and a normalized time near and prior to the phase with deepest SLP (Fig. 3b). The cyclone is thus deepening most strongly while propagating poleward. If only one forcing factor is high, either EGR (lower-right) or QG$\omega$ (upper-left), the cyclone seems to be in a period during its growth phase that favors near zonal (eastward) or even slight equatorward propagation. Interestingly, if both forcing factors are weak (lower-left corner), a

tendency for poleward propagation is discernible, although the propagation angle remains lower than for the case of combined strong forcing. One reason for this observation might be that cyclones which propagated poleward under strong dry-dynamic forcing continue to do so even after both forcings have vanished.

## 4.3 Cyclone-centered composites for poleward and eastward propagation

In the previous section, we saw that there is a general relationship between the deepening and the propagation of cyclones. Cyclones tend to propagate poleward when the deepening is strongest. In this section, we link this result back to the dry-dynamic forcing QG$\omega$ and EGR by means of cyclone-centered composites for cyclones that propagate predominantly poleward compared to eastward propagating cyclones.

Figure 10 shows the QG$\omega$ forcing (color) for poleward propagating cyclones (Fig. 10a) and for cyclones propagating more eastward (Fig. 10b). The categories are defined according to a range of propagation angles: angles between 35 and 65° for poleward propagation, and between -5° and 25° for eastward propagation. The black dot represents the SLP minimum (cyclone centre) and the black arrow in the center indicates the mean propagation direction of the cyclone within the following 6 h. To make the comparison easier, a white point indicates in both fields the position of the maximum QG$\omega$ forcing for poleward propagation and a white cross indicates the maximum in case of an eastward propagation.

In both cases, the cyclone center is located close to and just equatorward of the maximum QG$\omega$ forcing. The maximum forcing is more pronounced for poleward propagating cyclones with values exceeding -0.24 Pas$^{-1}$ (Fig. 10a). The QG$\omega$ maximum for eastward propagating cyclones reaches -0.16 Pas$^{-1}$ (Fig. 10b). The forcing in case of poleward propagation by QG$\omega$ is purely poleward and the direction of propagation is northeastward (black arrow in Fig. 10a). In contrast, for eastward propagating cyclones the maximum of QG$\omega$ forcing (white cross) is not only weaker but also located to the northeast of the cyclone center resulting in a more zonal direction of propagation. In summary, it seems that the weaker amplitude and eastward shifted QG$\omega$ center leads to a more zonal cyclone propagation, whereas a QG$\omega$ maximum to the north is able to deflect the cyclone path poleward.

The EGR environment for poleward and zonal cyclone propagation is shown in Fig. 10c and d. In case of poleward propagating cyclones (Fig. 10c), the displacement vector is orientated essentially normal to the EGR field, pointing towards lower EGR values. This indicates that cyclones propagate away from high EGR values, which are found in this case in the southwestern sector of the cyclone where one would expect the associated cold front to be located. On the other hand, in the case of eastward propagating cyclones (Fig. 10d), the displacement vector is also locally normal to the EGR isolines, but the large-scale EGR environment has a stronger zonal orientation compared to the poleward propagating case. It is, however, difficult to judge what the exact contribution by the EGR environment is to the cyclone's propagation.

## 5 Conclusions

The deepening and propagation of extratropical cyclones occurs within a remarkable wide range of environments. In this study, we analysed the environment during the cyclone growth period in terms of the dry-dynamic forcing, its variability

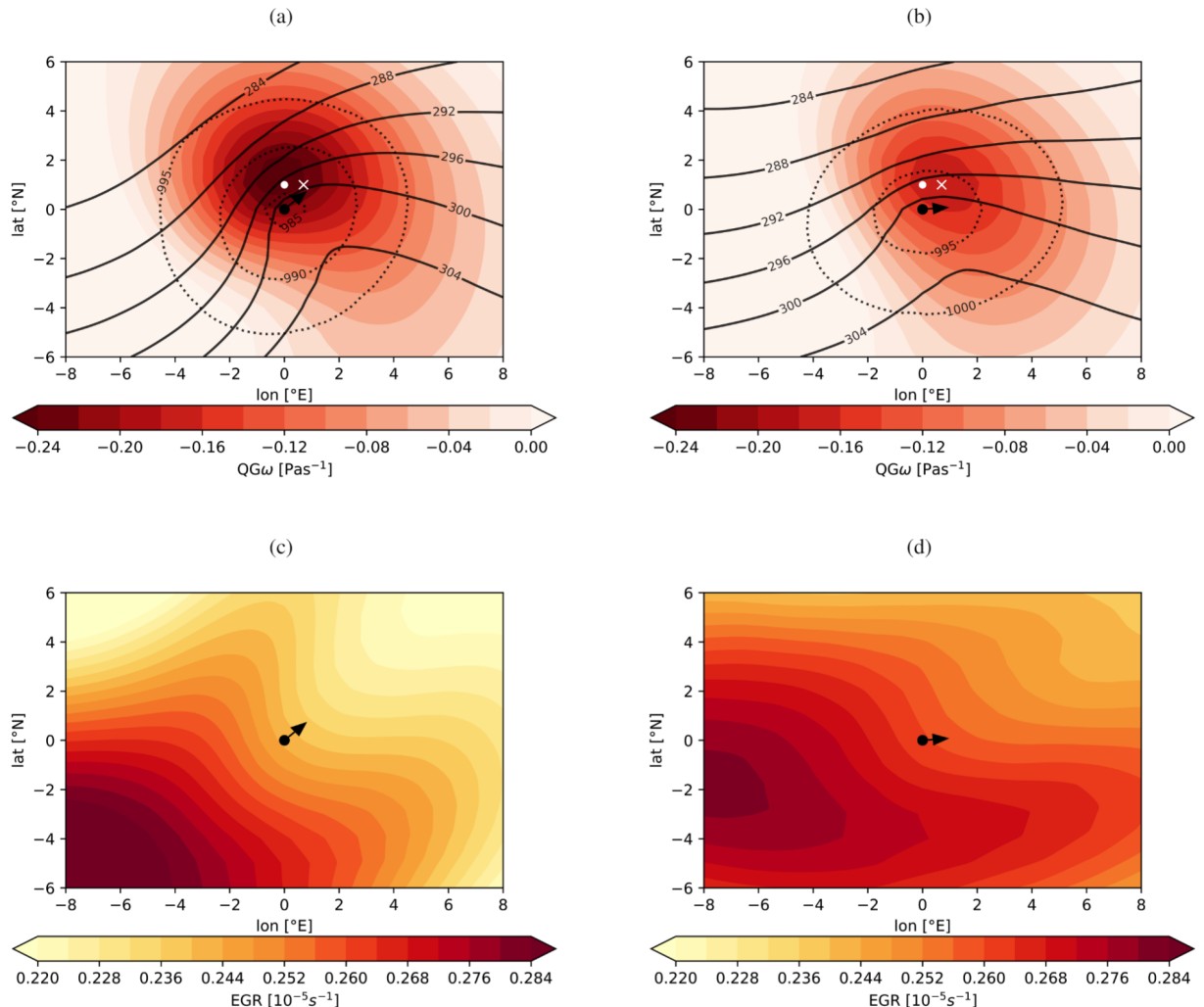

**Figure 10.** Cyclone-centered composites of QG$\omega$ (coloured contours in Pas$^{-1}$) for (a) poleward propagating cyclones ($35° < \alpha < 65°$) and (b) eastward propagating ($-5° < \alpha < 25°$) cyclones, and EGR composites (coloured contours in $10^{-5}$s$^{-1}$) for (c) poleward propagating cyclones and (d) eastward propagating cyclones. The black dot represents the cyclone center and the arrow the mean 6-h propagation direction of the cyclone. The white dot in (a) and (b) indicates the maximum of the QG$\omega$ forcing for poleward propagating cyclones and the white cross the QG$\omega$ maximum for eastward propagating cyclones, i.e., they mark the exact location of color-shaded QG$\omega$ fields. The white dot and cross are depicted both in (a) and (b) to highlight the spatial shift between the two forcing maxima. The black contour lines in (a) and (b) show $\theta_e$ (in K) at 850 hPa, the dotted lines SLP (in hPa).

and relationship with the cyclone propagation direction. To this aim, extratropical surface cyclones were identified and tracked
during the Northern Hemisphere cold season (October to March) based on six-hourly ERA-Interim data (1979-2016). Each time
step along every cyclone track was characterized in terms of its 12-hour deepening rate ($\Delta$SLP), the upper-level QG forcing

for ascent (QGω), the lower-tropospheric Eady Growth Rate (EGR) and the propagation direction. Since cyclone deepening and direction of propagation are determined by a cyclone's large-scale environment, the QGω and EGR forcing were averaged within a 1000 km radius around the cyclone center. To facilitate the comparison between the multitude of cyclone tracks, the phase of the cyclone evolution was quantified by introducing a normalized time axis, with -1 corresponding to genesis, 0 to the time instance of deepest sea-level pressure (SLP) and +1 to lysis. The analysis was restricted to the growth period of the life cycle between normalized times -1 and 0. The main results of the presented analysis can be summarized as follows:

- The largest 12-hour deepening rates result for a combination of high EGR and strong QGω forcing (category Q↑E↑), as expected and clearly visualized in the forcing histogram. For instance, a mean deepening rate of 6 hPa within 12 hours is found for a forcing of EGR= $1.3\,\text{day}^{-1}$ and QGω $= -0.22\,\text{Pa}^{-1}$. The second largest mean deepening rates are found for category Q↓E↑ (-3 hPa within 12 hours for EGR= $1.3\,\text{day}^{-1}$ and QGω $= 0\,\text{Pa}^{-1}$), followed by Q↓E↓ and Q↑E↓.

- An interesting asymmetry is discernible between conditions with high EGR and low QGω forcings and, conversely, low EGR and high QGω forcing: larger deepening rates result for the former than the latter. This indicates that baroclinic instability and substantial deepening rates can result even in situations with moderate upper-level (QGω) forcing as long as the EGR is high enough. The opposite is, however, not true: even substantial upper-level forcing results only in weak deepening rates if EGR is low.

- The four different forcing categories dominate different periods of the growth phase. The early phase closest to genesis is dominated by Q↓E↑, while thereafter increasing QGω forcing categories dominate. This indicates that the four forcing categories represent also different phases in the evolution of cyclones: with EGR-driven deepening occurring earlier in the life cycle than QGω-driven deepening.

- The flow situations in the environment of the four forcing categories significantly differ in terms of upper-level jet structure, QGω and EGR, as revealed by cyclone-centered composites. Category Q↓E↑ is characterized by a weakly disturbed, southwest-to-northeast oriented upper-level waveguide, which can also be taken as a proxy for a corresponding jet structure. In category Q↑E↑, the upper-level PV field is reminiscent of a strongly positive upper-level PV anomaly to the west of the surface cyclone, i.e., this situation shows the characteristic westward tilt with height for growing cyclones in a baroclinic atmosphere. In category Q↑E↓, the surface cyclone center and the upper-level PV maximum essentially coincide, indicative for a barotropic atmosphere with weaker deepening rates. Finally, for the category with weak forcing (Q↓E↓) only a weak upper-level PV structure is discernible. Of course, the QGω and EGR environments corresponding to these categories are in accordance with the expectation from the upper-level PV structures. The patterns show also how critical the mean forcing depends on the relative position to the cyclone center. In particular, we note that the surface cyclone is not located at the location of maximum EGR and/or QGω, but at locations with strong gradients.

- During phases of strong deepening, cyclones show a tendency to propagate poleward. This behaviour scales with the degree of deepening: The propagation shifts from an eastward to a more northeastward or poleward orientation as the

12-hour ΔSLP deepening becomes stronger. As poleward propagating cyclones are associated with stronger deepening, they also show a higher dry-dynamic forcing in their environment, i.e., poleward propagation is found in category Q↑E↑. Thereby, the cyclone propagates in the direction of maximum QGω forcing, as revealed by the cyclone-centered composites. In contrast, eastward propagating cyclones are associated with weaker deepening and lower EGR and QGω forcing. They are directed to a more eastward propagation by the QGω forcing.

- Geographically, there are distinct regions that specifically show a clear tendency for poleward propagation, while other regions are associated with a more equatorward-oriented propagation direction. The regions with strong poleward propagation coincide with areas of maximum eddy kinetic energy (the North Atlantic and North Pacific storm tracks), supporting the interplay between propagation direction and cyclone strength. Eastward or even equatorward propagation is predominant, e.g., to the lee of the Rocky Mountains or the Himalayas.

- Overall, remarkably distinct geographical patterns, with only weak spatial overlap, emerge from the four forcing categories. Category Q↑E↑ is characteristic for the entrance regions of the North Atlantic and North Pacific storm tracks; category Q↓E↑ is typical over continental North America, along the southern tip of Greenland and over East Asia and partly also over the western North Pacific; category Q↑E↓ is predominant in a zonal band around 30° N extending off the U.S. west coast across the North Atlantic (with peak frequencies) to the eastern Mediterranean; finally, the category with weak forcing (Q↓E↓) has is local maximum over the North Atlantic, further north than Q↑E↓ and further south than Q↑E↑.

This study is strongly based on EGR and QGω as two key drivers for cyclone development. This choice reflects a very classical view on extratropical cyclone growth, and it is worthwhile to briefly discuss our results in this perspective. The low-level Eady growth rate is a measure for the strength of the low-level baroclinicity and for the growth rate potential of the most rapidly growing wave (Lindzen and Farrel 1980). It is proportional to the baroclinic conversion rate if multiplied by the eddy heat flux (e.g., Schemm and Rivière (2019)). However, high EGR alone gives only an idea of growth potential. In order for actual cyclone growth to occur, the baroclinic instability must be released by a disturbance, for example, through an upper-level trough that approaches the zone of enhanced low-level baroclinicity from upstream (for example as for the Petterssen and Smebye (1971) type B cyclone or as depicted from a PV perspective in Fig. 21 of Hoskins et al. (1985)). The vertical lifting ahead of this upper-level trough acts as the required trigger for baroclinic growth, and it is the strength of the vertical lifting that we quantify through the QG omega equation. It is the combination of both that matters and we must expect strong growth if QG omega and Eady growth rate are both strong. In this sense, the results of this study are, in the broadest sense, a climatological confirmation of existing knowledge in the field of dynamic meteorology. However, the presented climatology clearly shows how great the regional variability of the two variables can be during cyclone growth and in which broad spectrum similar cyclone growth can occur.

The EGR/QGω-based perspective is closely related to the PV-based perspectives on cyclone growth, though both are not identical. For example, the upper-level trough that induces the QGω forcing can equivalently be regarded as an upper-level PV anomaly that induces, based on PV invertibility, a cyclonic circulation over the low-level zone of baroclinicity (Hoskins

et al., 1985). The surface warm air anomaly that forms the cyclone warm center, in turn, can be regarded as a low-level (or even surface) PV anomaly. Both PV anomalies mutually amplify and accelerate the cyclonic circulation and poleward heat transport. PV provides via the invertibility principal information on horizontal circulation, temperature, and pressure anomalies. However, the forced vertical motion or the strength of the baroclinicity cannot readily be deduced from PV alone, and we therefore built this study on EGR and QG$\omega$, because both allow a direct quantification of these two drivers. Still, the cyclone-centered composites presented in the study overlaid with upper-level PV highlight the close relationship between the two perspectives. Further, although not discussed in detail, intense low-level baroclinic zones (fronts) typically are accompanied by high-PV bands (Schär and Davies, 1990; Joly and Thorpe, 1990; Schemm and Sprenger, 2015). In summary, both perspectives should be regarded as interconnected.

The study comes with some caveats, but also some rewarding ideas for further research. While the produced climatology covers 38 years of data (ERA-Interim) and covers the whole Northern Hemisphere, a more refined analysis of specific regions might be worthwhile. For instance, it would be interesting to see how cyclones in the Mediterranean split between the different forcing categories; or if cyclones in the Southern Hemisphere also exhibit such a clear geographical split as is found for the Northern Hemisphere. Furthermore, an interesting extension would be not to split a cyclone track, as is done in this study, into separate and independent short-term segments, but to consider the whole evolution of a cyclone (including its decaying phase) as an evolution path in the 2D forcing diagram and see to which degree the propagation and pressure evolution of a cyclone can be understood by it. This, however, would ask for a refined definition of the QG$\omega$ and EGR environments of a cyclone, which is not only restricted to the mean values within a 1000 km radius but would also take into account the spatial and cyclone-relative position of the forcing factors. It would also be a most rewarding extension of this study to incorporate diabatic forcing factors in the analysis, i.e., leave the realm of dry dynamics and expand the forcing diagrams into three dimensions. Finally, the combined EGR and QG$\omega$ perspective could be used to identify biases in the representation of cyclone dynamics and storm tracks in present-day and historic CMIP6 simulations (Priestley et al., 2020). For instance, midlatitude storm tracks are known to be too zonal in historical CMIP6 simulations, and the robust EGR/QG$\omega$ diagnostic could readily be applied to temporally coarser dataset than the six-hourly ones used in this study.

*Data availability.* The ERA-Interim cyclone tracks and the EGR fields are available at a monthly resolution from the web page linked to Sprenger et al. (2017). Higher temporal resolution data (cyclone tracks, EGR) and QG$\omega$ fields can be obtained from M. Sprenger on request.

*Author contributions.* All authors contributed to the concept of the study. Most of the analysis was performed by P. Besson; the cyclone tracks, diagnostic QG$\omega$ and EGR fields were calculated by M. Sprenger; S. Schemm and M. Sprenger contributed with several diagnostic tools (e.g., cyclone-centered composites) and supported the interpretation, L. Fischer inspired the visualizations and helped with the statistical analysis. All authors contributed to the writing of the manuscript.

*Competing interests.* None of the authors has any competing interest

*Acknowledgements.* We would like thank MeteoSwiss for providing access to the ERA-Interim dataset. Thanks also to Heini Wernli for
530 helpful discussions on cyclone evolution, and to Timo Schmid whose excellent Bachelor thesis gave the inspiration for the 2D forcing
diagrams.

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
