# Peer review of "A global analysis of the dry-dynamic forcing during cyclone growth and propagation"

_Weather and Climate Dynamics, 2021_

## Referee Comment (RC2)

**Review of "A global analysis of the dry-dynamic forcing during cyclone growth and propagation" by Besson et al.**

https://doi.org/10.5194/wcd-2021-17

This manuscript investigates how dry dynamical factors influence the intensification of extra-tropical cyclones and their propagation direction. The analysis is based on 38 years of ERA-Interim reanalysis and focuses on the extended cold season. The analysis is novel and even if some results are as expected, this manuscript is a valuable contribution to the field. However, there are two major issues with the manuscript which need to be carefully considered before this manuscript can be accepted. First, a theoretical background and justification of the two variables is lacking (major points 1) as is a clear explanation of all diagnostics and how they were computed (major point 2, minor point 3). Secondly, I am concerned about how some of these results may depend on subjective choices made in this analysis, namely the size of the bins in the phase space (major point 3) and on the decision to analyse all times between the time of genesis and minimum mean sea level pressure together (major point 4). In addition, minor comments – which also often refer to subjective choices made in the analysis - are also detailed below.

**Major comments**

1. The choice of the two variables (Eddy Growth Rate and the QG upper-level forcing) is not clearly explained / motivated. Why these two specific variables in the specific layers and not other variables / different layers? The manuscript also lacks an in-depth theoretical discussion about what these two variables really represent. In particular, it is stated that the Eddy Growth Rate represents the low-level baroclinicity, which is true, but the lower tropospheric stability (N) also has a large effect on the Eady Growth Rate. This aspect is not considered in the analysis and interpretation of the results. Lastly, how the two variables relate to each other is not considered either theoretically or in the analysis. It would be very interesting to see a map of how these two variables correlate with each other in a climatological sense (without the additional requirement of a cyclone being present). This could be shown in a third panel in Figure S2.

2. Related to major point 1, additional details should be presented in this manuscript concerning how these diagnostics were calculated rather than just referring to Graf et al (2017). It is not clear over which layers the Eady Growth Rate is calculated – in line 86 it is stated that the Eady Growth Rate is "representative for low-to-mid tropospheric levels". Additional aspects that need to be considered are: are the vertical derivatives calculated taking just two pressure levels? How is the static stability in the omega equation calculated (often this is taken to be a global constant)? How is the Brunt-Väisälä frequency calculated? - the equation is given in terms of height not pressure. Is model level data from ERA-Interim used (as suggested in line 80) or is it pressure level data?

3. Line 144 / Figure S1 / Lines 160-161. Selection of the number and width of bins in the 2D histogram. What is the justification for using linearly spaced bins? Would the results differ if the bins were designed so that each bin at approximately the same number of samples present? This is potentially a critical problem in this analysis and needs to be investigated. In

the current format, the 4 forcing categories defined in ~Line 160 and used for much of the analysis have hugely different number of data points (Figure S1). For example, the 4 boxes in the top left (Q↑E↓ ) have 1675 data points whereas the 4 boxes in the bottom left (Q↓E↓ ) have 21,042 data points – more than 10 times as many.

4. Lines 135-136 and many of the results. It is stated that "All of the analysis in section 3 and 4 will be restricted to the phase with normalized times between -1 and 0". I agree it makes sense to focus on the time during which the cyclones are intensifying, however, I do not agree that considering all times between the time of genesis and time of minimum mean sea level pressure all together is a good decision. This is because cyclone structure, location (geographically and relative to the jet) and clearly intensify varies hugely during this time. Because of this, in most previous cyclone composite studies (e.g. Catto et al, 2010, Dacre et al, 2012, Flaounas et al, 2015) different offset times relative to the time of maximum intensity are considered separately. For example, how do Figures 4 and 5 change if only normalised times from e.g. -1 to -0.5 or from -0.5 to 0 are considered? Also, would the numbers of timesteps in each bin (e.g. Figure S1) change if the time period was split into two?

**Minor comments**

1. Line 100, The radius of 1000 km is somewhat arbitrary but I appreciate that some value needs to be selected. Some brief justification is necessary though e.g. has a similar radius been used by others? More importantly, it should be clearly stated what this radius is meant to represent -  the size of the cyclone or the size of the area which can affect the subsequent evolution of the cyclone?

2. Line 138. The three geographic boxes. Are the results sensitive to the choice of these areas? The reason for this comment is that these areas are quite large and cover the central and end parts of the main storm tracks regions. Why not consider the start and end of the storm track regions separately as it is well known that cyclones with their genesis in the western North Atlantic differ from those with their genesis in the eastern North Atlantic.

3. Section 2.3 / Figure 1. The 12-hour change in SLP. This diagnostic is not clear to me. At first I thought this was the maximum 12 hour deepening rate (the text on line 147 caused this thought) but this cannot be correct given that it has positive and negative values in the histograms in Figure 1. Please can it be clarified what this is – this aspect caused me quite a lot of confusion throughout the manuscript.

4. Related to minor point 3, how does the 12-hour change in mean sea level pressure relate to the normalised time presented in lines ~125 – 135? e.g. How does this "real" time relate to normalised time?

5. Lines 154-155. Does this statement that none of the distributions is strongly skewed apply to all areas of the phase space e.g. what would the distributions look like for a point in the middle which has a much larger sample size? Potentially, the distributions become more Gaussian in the middle are are most extreme in the corners of the phase space?

6. Line 194 – Very minor comment. It is not clear (to me) what is meant by the Greenland Shelf – is this the land / ice mass of Greenland?

**Figure comments**

- Figure 2 caption. Can the months analysed be added here? It would help remind a reader that only the extended cold season is analysed.

- Figure 4 and 5. These are presented on a cyclone-centre relative grid in terms of longitude and latitude. How does the 1000 km radius used earlier relate to this lon / lat space? The physical distance in kilometres between 20 degrees of longitude decreases with increasing latitude. Is it valid to assume that 10 degrees in longitude or latitude is approximately 1000 km? If valid, could this assumption be added to the captions?

- Figure 4 and 5 – related to major points 3 and 4 above. The mean values are presented in these cyclone composites but how much variability is there within each of these composites? This variability may be large due to the different offset times considered all together here. Furthermore, the variability may differ considerably between the 4 classes given the difference in the number of samples in each class.

- Figure 7. What are the grey / blocked out areas in the bottom left and bottom right of this figure? Also the caption needs a capital letter for "additionally".

- Figure S2 has a odd map projection and is lacking longitude and latitude labels. Can this projection / figure style be changed to match the maps shown in the main manuscript?

**References**

Catto, J.L., Shaffrey, L.C. and Hodges, K.I., 2010. Can climate models capture the structure of extratropical cyclones?. *Journal of Climate*, *23*(7), pp.1621-1635.

Dacre, H.F., Hawcroft, M.K., Stringer, M.A. and Hodges, K.I., 2012. An extratropical cyclone atlas: A tool for illustrating cyclone structure and evolution characteristics. *Bulletin of the American Meteorological Society*, *93*(10), pp.1497-1502.

Flaounas, E., Raveh-Rubin, S., Wernli, H., Drobinski, P. and Bastin, S., 2015. The dynamical structure of intense Mediterranean cyclones. *Climate Dynamics*, *44*(9-10), pp.2411-2427.

---

## Author Response (AR1)

**Reviewer 1**

In this manuscript the authors investigate the dry-dynamics forcing influencing extratropical cyclones growth and propagation direction. Specifically, this is done by analyzing the Eady Growth Rate (EGR) and the QGw forcing along the cyclone tracks. Overall, this is a well-written paper that presents interesting results. However, I was missing some more theoretical basis and motivation for the choice if these two diagnostics. Most of the conclusions of this study are not really surprising (and generally agree with previous studies), but the paper is still interesting and deserves publication after some revision. My comments are given in more detail below.

**Main comments:**

1) I am not sure what is the benefit of using the two diagnostics above (namely, EGR and QGw), as opposed to, for example, a PV tendency equation. I was missing a more theoretical basis and motivation for the choice if these two diagnostics. Also, how do these relate to terms appearing in the PV tendency equation? For example, for the QGw term, is this essentially the vertical advection wdq/dt? For the EGR term, I think you should discuss more what it represents and how it relates to earlier studies (e.g., from a PV perspective). What is the equation in which EGR enters as a forcing term? Does this term essentially represents the induced meridional advection by the upper level PV? It seems to me like less can be learned on the actual dynamics by looking at these two diagnostics alone. So I think you should at least elaborate on your choices for diagnostics in the introduction and methods, and which equation motivates your analysis.

**Reply:** Thank you for suggesting a clarification concerning this choice. The Eady growth rate and the QG omega equation comprise a very classical view on extratropical cyclone growth. The low-level Eady growth rate is a measure for the low-level baroclinicity and for the growth rate of the most rapidly growing wave (Lindzen and Farrel 1980). The instability of the baroclinic zone is however only released if it is triggered by a perturbation, for example, an upper-level trough approaching the zone of low-level of baroclinicity (Petterssen and Smebye's type B cyclone; 1971). The vertical lifting ahead of the upper-level trough can act as the required trigger for baroclinic growth and the strength of vertical lifting is quantified through the QG omega equation. The Eady growth rate quantifies the strength of the low-level zone of baroclinicity. Thus, one would expect strong growth if QG omega and Eady growth rate are both strong, confirming numerous case studies. Here, we present a climatological analysis.

The Eady growth rate enters, after multiplication with the meridional eddy heat flux, the eddy available potential energy equation as part of the baroclinic growth term (Eq. 4 in Schemm and Rivière 2019). This equation shows that baroclinic growth is proportional to both, the eddy heat flux and the baroclinicity, measured here by the Eady growth rate.

PV is an elegant and unifying but different concept to understand the same cyclone dynamics. There are some links, for example, the upper-level trough is seen as an upper-level PV anomaly that induces a cyclonic circulation over the low-level zone of baroclinicity (Hoskins et al. 1985). The surface warm air anomaly that forms the cyclone warm center is then seen as a low-level PV anomaly. Both PV anomalies mutually amplify and accelerate the cyclonic circulation and poleward heat transport. PV provides via the invertibility principal information on horizontal circulation, temperature, and pressure anomalies. However, the forced vertical motion or the strength of the baroclinicity cannot be deduced from PV. There are some overlaps between both perspectives, for example, the Eady

growth rate is large along the cold front of a cyclone, which corresponds to an elongated band of positive PV. We agree with the reviewer that the forced vertical motion enters the eddy-decomposed PV equation via the vertical advection (w'dq'/dz). We will implement the above discussion into the revised manuscript.

2) Section 3.1: perhaps you should show first density maps of along track E and Q regions separately (or do these just look very similar to the climatological fields given in the supplementary information?).

Reply: Thank you for this input. For the most part, the separate distributions of Q and E are a combination of the category forcing distributions. For instance, E↓ (lower left in Replies Fig. 1) combines the distributions of Q↑E↓ and Q↓E↓. For E↑ (lower right in Replies Fig. 1) this also works well, as we see all the same areas represented as in the forcing categories. However, for Q↑ (upper right in Replies Fig. 1) the major hot spot in the Atlantic off the coast of Northwestern Africa disappears when we only distinguish Q by itself instead of considering the geographical density distributions of the forcing categories. Overall, we therefore believe it is sensible to keep the geographical desnitsity distributions forcing category in the article.

[Figure]

Replies Figure 1. Geographical density distributions of Q and E separately. Top row Q and bottom row E values, low ↓ on the left, high ↑ on the right.

3) Fig. 2a: The high Q and low E that occur at the downstream regions of the storm tracks- perhaps these are related to secondary (downstream) cyclogensis?

Reply: Downstream region are very likely related to secondary cyclogenesis; we added a note.

4) Section 3.2: You can just plot the averaged Q and E as a function of normalized time to support these findings.

[Figure]

**Replies Figure 2.** Evolution of QG omega and EGR along normalized time, from cyclone genesis to decay.

**Reply:** The evolution of QG omega and EGR according the normalised time is shown in the figure, whereby it has to be kept in mind that the evolution does not necessarily reflect the real temporal evolution of a cyclone life cycle. The figure only shows, for instance, that at the time of genesis (at time -1) cyclones are associated with intermediate EGR values and high QG omega values. This seems physically plausible because a low-level baroclinic zone (as expressed with EGR) is only one ingredient for cyclogenesis. The upper-level forcing (QG omega) might act as a trigger to release the baroclinic instability, and hence allow for the further cyclone deepening. Interestingly, immediately after genesis, e.g. at normalised time -0.8, the EGR values become larger and the QG omega values considerably weaker. This agrees with our statement that the cyclone deepening is governed in the early phase of the growth period by the high EGR values. Only later, towards the phase of deepest SLP and when the cyclone has attained a mature state, the upper-level forcing becomes large again. The steady increase in QG omega between normalised times -0.8 and -0.2 thereby reflects the co-evolution of the near surface and the upper-level flow. Finally, near normalised time 0, both forcing factors steeply decrease, which -- of course -- makes sense since the cyclone has already reached its mature stage and starts to decay for times larger than 0.

5) Would diabatically forced cyclones enter the low Q and low E group? If so, I would still expect to find high deepening rates for this cluster.

**Reply:** We agree with the reviewer, it is not clear if the category contains a high fraction of diabatically driven cyclones. The corresponding remark was removed.

6) The fact that the regions of maximum deepening in the two oceanic basins coincide with local maxima in mean poleward propagation angles in not surprising, and entirely agrees with previous studies that showed the role of the westward tilt (through induced meridional advection) and diabatic heating to the poleward propagation (e.g., Rivière et al. 2012, Coronel et al., 2015; Tamarin and Kaspi, 2016).

**Reply:** Yes, and all three publications are already cited in the introduction. We added these once more in the corresponding result section 4.1 and the discussion of Fig. 7.

7) Fig. 9d: It seems like the jet in this case is contributing more to zonal advection.

**Reply:** The composite displays an overall more zonal flow situation, which agrees with a more zonal

jet. The strongest upper-level forcing is found downstream of the low-pressure system with only little to no poleward displacement.

8) As you state: "it is difficult to judge what the exact contribution by the EGR environment is to the cyclone's propagation". I think this is because the EGR measure mixes the zonal (through U) and meridional (through the induced advection) influences. On the one hand, I would expect more poleward propagation where EGR is high (since this implies westward tilt and hence poleward advection by the upper level PV), but high EGR also implies strong zonal advection. This is why I think the EGR diagnostic is not very helpful for studying the propagation angles.

Reply: Yes, we agree, we also find that the EGR provides only limited insight into the direction of propagation. Note that we use the wind magnitude and not the zonal wind component in the computation of EGR. We will add the equation to the manuscript.

The motivation behind using the EGR was primarily not the study of the direction of propagation, but the link with the observed growth rate since EGR is a theoretical measure for the growth potential. As such, it does not by definition provide information on the horizontal circulation, although we may assume that there is strong meridional advection in case of strong growth. However, as shown in this study, it is the combination of high upper-level forcing of vertical motion and low-level EGR that is – as expected - associated with strong growth. PV is a more convenient framework for the study of the propagation direction, though it does not allow for a quantification of vertical motion (only if connected to the QG omega equation).

9) QGw influence on propagation (the results you describe when discussing Fig.9a,b)- can you explain why this is what we find? what is the underlying mechanism

Reply: The underlying mechanism is low surface pressure that is due to vertical motion and upper-level divergence. The upper-level location of the maximum in forced vertical motion (which occurs ahead of the trough) thus drives the following change in location of the surface low pressure. As such the QGw can function as guiding for cyclone propagation.

10) In general, throughout the paper, I was missing a discussion (beyond describing your results) of what the underlying mechanisms are.

As outlined in above responses we will add more discussion on underlying mechanisms in to the revised manuscript.

**Minor comments:**

1) Line 47: turns=turn Reply: Thank you, this has been changed.

2) Line 70: Wrong citation here. The relevant citation here should be: T. Tamarin-Brodsky and Y. Kaspi, "Enhanced poleward propagation of storms under climate change", Nat. Geosci., 10.1038/s41561-017-0001-8 (2017).

Reply: Thank you, this will be addressed for the revised manuscript.

3) Line 335: I think you meant Fig. 9a. Reply: Thank you, this has been changed.

**Reviewer 2**

This manuscript investigates how dry dynamical factors influence the intensification of extra-tropical cyclones and their propagation direction. The analysis is based on 38 years of ERA-Interim reanalysis and focuses on the extended cold season. The analysis is novel and even if some results are as expected, this manuscript is a valuable contribution to the field. However, there are two major issues with the manuscript which need to be carefully considered before this manuscript can be accepted. First, a theoretical background and justification of the two variables is lacking (major points 1) as is a clear explanation of all diagnostics and how they were computed (major point 2, minor point 3). Secondly, I am concerned about how some of these results may depend on subjective choices made in this analysis, namely the size of the bins in the phase space (major point 3) and on the decision to analyse all times between the time of genesis and minimum mean sea level pressure together (major point 4). In addition, minor comments – which also often refer to subjective choices made in the analysis - are also detailed below.

**Major comments**

1) The choice of the two variables (Eddy Growth Rate and the QG upper-level forcing) is not clearly explained / motivated. Why these two specific variables in the specific layers and no other variables / different layers? The manuscript also lacks an in-depth theoretical discussion about what these two variables really represent. In particular, it is stated that the Eddy Growth Rate represents the low-level baroclinicity, which is true, but the lower tropospheric stability (N) also has a large effect on the Eady Growth Rate. This aspect is not considered in the analysis and interpretation of the results. Lastly, how the two variables relate to each other is not considered either theoretically or in the analysis. It would be very interesting to see a map of how these two variables correlate with each other in a climatological sense (without the additional requirement of a cyclone being present). This could be shown in a third panel in Figure S2.

Reply: Thank you for suggesting a clarification concerning this variable choice. We will add a more detailed discussion to the introduction. The Eady growth rate is a measure for the strength of the baroclinicity that is defined as the meridional temperature gradient divided by the static stability, which enters the Eady growth rate in the denominator. So, we agree if N is low, the Eady growth rate is large (and vice versa). EGR is therefore a measure for the growth potential of the most rapidly growing wave (Lindzen and Farrel 1980). However, to initiate baroclinic growth a trigger is needed, for example, in the form of vertical lifting ahead of an upper-level trough. This forced vertical lifting is quantified by the QG omega equation. Hence, both together allow to quantify the two-key ingredients in the traditional view (Petterssen and Smebye's type B (1971)) on cyclone dynamics: The strength of the low-level zone of baroclinicity and the strength of the upper-level triggering.

Good point about the climatological relationship between QG omega nd EGR, we will add the following map (Replies Figure 3 (a)) (in a projection consistent with the other figures in the manuscript) of the correlation between EGR and QG omega in the revised supplement as panel (d) in Figure S2. It shows, as suggested by the reviewer, the linear correlation coefficient between EGR and QG omega, whereby all timesteps for the extended winter (Oct-Mar) between 1980-2016 are included in the calculation. The Figure S2 in the supplement will contain three subpanels: (a) correlation between EGR and QG omega; (b) EGR, and (c) QG omega

(a) linear correlation between QG omega and EGR

(b) EGR

(c) QG omega

[Figure]

**Replies Figure 3.** Climatological EGR (b), QG omega(c), and their correlation (a), in extended winter.

It is instructive to relate this diagram to Figure 2 of the manuscript, where the geographical distribution of the four forcing categories are shown. For easier reference, we copy this figure here into the reply document and then briefly discuss some aspects:

[Figure]

**Replies Figure 4.** Copy of Figure 2 in article.

Category Q↑E↑ (large negative QG omega and large positive EGR values) is found at the beginning of the North Atlantic and Pacific storm tracks. These are regions where the correlation between QG omega and is negative and also somewhat enhanced compared to mid- and east-oceanic regions, i.e., the correlation matches with expectation. The link to the other categories is not particularly strong. There is an indication that category Q↓E↑ (small or even positive QG omega and large positive EGR) over North America goes along with weak or even positive correlations, i.e. they are counteracting. Overall, these 'matches' between the category forcing and the correlation map are weak, and thus reflect a more complex interplay (or lack of linear correlation) between the two factors.

With respect to the correlation between EGR and QG omega, irrespective of the four forcing categories, the following conclusions can be drawn: (i) the correlation remain rather weak in the main North Atlantic and Pacific storm tracks; (ii) the correlation is somewhat larger in the western (entrance) part of the storm tracks, and becomes smaller towards the east; (iii) the largest anti-correlation are found in the subtropics, to the east of China, and to the west of North America and Africa; (iv) positive correlations are essentially restricted to the region to the east of the Himalayas. Note that negative correlations indicate that the two forcing factors potentially act together in a cyclone's deepening, as positive EGR rates (baroclinic forcing) coincides with negative QG omega values (forcing for lifting). Interestingly, to the east of the Himalayas, with a positive correlation coefficient, the two facts seem to counteract with respect to cyclone deepening.

In addition to the linear correlation between QG omega and EGR, as in Figure 2, we will also show as panels (a)-(b) of an additional Figure S3 the winter climatologies of the negative part of QG omega (a) and the positive part of QG omega (b). Whereas panel (a), i.e., the negative part of QG omega, has already been shown in the initial version of the manuscript, the positive counterpart in panel (b) is new. We think that it is worthwhile to show the positive and negative part separately, to avoid cancellation in a climatology of QG omega due to the typical co-occurrence of positive and negative poles. The two panels are:

(a) QG omega (negative part)

[Figure]

(b) QG omega (positive part)

[Figure]

**Replies Figure 5.** Climatological QG omega negative part (a), and QG omega positive part (b), in extended winter.

The general patterns are similar in panels (a) and (b), which underlines the fact that the positive and negative anomalies often co-occur in QG omega dipoles. Still, there are noteworthy local differences. As one specific example, the positive pole in the eastern Mediterranean is located somewhat further to the west compared the negative anomaly. This, most likely, indicates that the positive and negative anomalies are part (as poles) of common weather systems, e.g., to the west and east of a short-wave trough.

2) Related to major point 1, additional details should be presented in this manuscript concerning how these diagnostics were calculated rather than just referring to Graf et al (2017). It is not clear over which layers the Eady Growth Rate is calculated – in line 86 it is stated that the Eddy Growth Rate is "representative for low-to-mid tropospheric levels". Additional aspects that need to be considered are: are the vertical derivatives calculated taking just two pressure levels? How is the static stability in the omega equation calculated (often this is taken to be a global constant)? How is the Brunt-Väisälä frequency calculated? - the equation is given in terms of height not pressure. Is model level data from ERA-Interim used (as suggested in line 80) or is it pressure level data?

**Reply:** Thank you for pointing to this lack in technical details. We fully agree with the reviewer that the description in the initial submission has to be improved. In particular, we will add the following additional pieces of information to the text:

- EGR: The Eady Growth Rate represents the layer between 850 and 500 hPa and vertical pressure

derivatives are accordingly calculated as finite differences between these two levels. The corresponding discretized equation for EGR looks as follows (copied from Graf et al., 2015):

$$EADY = 0.31 \cdot \frac{f}{N_{500-850}} \cdot \left[ \left( \frac{u_{500} - u_{850}}{Z_{500} - Z_{850}} \right)^2 + \left( \frac{v_{500} - v_{850}}{Z_{500} - Z_{850}} \right)^2 \right]^{1/2}$$

Here, $N_{500-800}$ represents the pressure-weighted average of the Brunt-Väisälä frequency between 500 and 850 hPa. N itself is calculated on ERA-Interim model levels, and the vertical average is subsequently calculated. We added the discretized equation to the manuscript, such that the calculation method for this key parameter of the study is perfectly clear.

- QGomega: A detailed description of the inversion method is, of course, beyond the scope of this manuscript. We add a reference to a technical report (Reinert, 2009; in German) to be also perfectly clear how numerically the inversion is done and how the different forcing terms, the static stability and all needed interpolations/derivatives (and other numerical steps) are calculated. Although the technical report is in German, it's mathematical nature should allow one to easily grasp the essential steps. The report also includes some sensitivity studies and discusses some case studies. To the specific point addressed by the reviewer, i.e., the static stability used in the Omega equation: The vertical stability is not constant in the domain, but a 1D vertical profile is used instead. This 1D basic state is calculated as the domain avaerge of static stabilities, i.e. sigma(z) is the horizontal average over sigma(x,y,z). This information is now added to the manuscript. Further, we add the equation that is used to calculate the static stbility, which in the afore-mentioned technical reads as follows:

$$\sigma = -\frac{RT_v}{p} \frac{d \ln(\theta)}{dp} = -\frac{R}{p} \frac{T_v}{\theta} \frac{d\theta}{dp} \ ,$$

Reference: Reinert, P., 2009: Bericht zum Programm zur Berechnnug der diagnostischen quasigeostrophischen Vertikalgeschwindigkeit. Technischer Bericht, 15 pp. [available on request from the authors].

3) Line 144 / Figure S1 / Lines 160-161. Selection of the number and width of bins in the 2D histogram. What is the justification for using linearly spaced bins? Would the results differ if the bins were designed so that each bin at approximately the same number of samples present? This is potentially a critical problem in this analysis and needs to be investigated. In the current format, the 4 forcing categories defined in ~Line 160 and used for much of the analysis have hugely different number of data points (Figure S1). For example, the 4 boxes in the top left (Q↑E↓) have 1675 data points whereas the 4 boxes in the bottom left (Q↓E↓) have 21,042 data points – more than 10 times as many.

[Figure]

**Replies Figure 6.** 2D forcing histograms with EGR (in day−1) on the x-axis and QGω (in Pa s−1) on the y-axis. On the left the Number of ΔSLP divided by the normalized time values within each 2D bin is colored, with darker purple-blue colors indicating a smaller bin width. On the right the mean of the 12-hour ΔSLP distribution within each 2D bin is colored, with darker red colors indicating a stronger cyclone growth.

**Reply:** This is a good point. To address this issue, we have performed an initial sensitivity analysis, where the bins in the middle section (which have the most values) are further divided into smaller bins. The amount of values in the smaller bins are now similar to the rest of the forcing histogram (left panel in Replies Figure 6). The mean dSLP values in the smaller bins, as seen in the panel on the right, are still similar to the "normal-sized" ones. The pattern in that middle area does change slightly. For instance, the lower right corner shows a stronger negative dSLP tendency in the smaller bins, and conversely, the upper right corner dSLP values are less negative in the smaller bins. However, the general big picture is not affected to a significant degree by the changes in bin size. The relevant pattern remains.

4) Lines 135-136 and many of the results. It is stated that "All of the analysis in section 3 and 4 will be restricted to the phase with normalized times between -1 and 0". I agree it makes sense to focus on the time during which the cyclones are intensifying, however, I do not agree that considering all times between the time of genesis and time of minimum mean sea level pressure all together is a good decision. This is because cyclone structure, location (geographically and relative to the jet) and clearly intensify varies hugely during this time. Because of this, in most previous cyclone composite studies (e.g. Catto et al, 2010, Dacre et al, 2012, Flaounas et al, 2015) different offset times relative to the time of maximum intensity are considered separately. For example, how do Figures 4 and 5 change if only normalised times from e.g. -1 to -0.5 or from -0.5 to 0 are considered? Also, would the numbers of timesteps in each bin (e.g. Figure S1) change if the time period was split into two?

**Reply:** We agree that it is a sensible idea to also look at separate parts of the intensification phase also. In terms of QGomega, we do not see drastic differences between the two parts of the

intensification phase (-1 to 0.5, 0.5 to 0). The structure of the upper level PV changes slightly and usually results in a larger PV gradient around the cyclone centre, as can be seen in the following figure. The Q↑E↓ composite shows a slightly stronger maximum QGω in the later stage of the intensification. Furthermore, the upper level PV gradient strengthens as well as the maximum PV, which vertically aligns with the cyclone centre. For EGR, there is no discernible difference between the two stages. The QGω in the Q↑E↑ composite does not change in terms of strength, and the cyclone centre remains in the region of maximum QGω. The upper level PV develops a stronger gradient in the second phase of intensification. The maximum PV is again over the cyclone centre. While the cyclone centre is close to the region of maximum EGR in the first stage, it "moves" further away in the second stage. This is not surprising as cyclones reduce baroclinicity. Both composites including Q↓ (two figures in the bottom row) show similar QGω structures and values in either stage of intensification. The position of the cyclone centre with respect to the region of negative QGω also does not vary a lot. While the EGR in the Q↓E↓ composites does not significantly differ, we see again a reduction in EGR in the Q↓E↑ from the first to the second phase. To summarise, we see significant changes in upper level PV in the Q↑ composites and in EGR in the E↑ composites.

[Figure]

**Replies Figure 7.** Cyclone-centered composites of contour lines of upper-level PV (in pvu) on the 320 K isentrope (black dashed contours), QG omega (red to blue shading), and EGR (yellow to red shading), Grey arrows represent the windfield at 300hPa, during different parts of the growth phase (−1 < $t_{norm}$ < -0.5, and −0.5 < $t_{norm}$ < 0). One set of four panels for the four forcing categories: (top left) Q↑E↓, (top right) Q↑E↑, (botom left) Q↓E↓ and (bottom right) Q↓E↑. The black dot represents the cyclone centre (SLP minimum).

**Minor comments**

1) Line 100, The radius of 1000 km is somewhat arbitrary but I appreciate that some value needs to be selected. Some brief justification is necessary though e.g. has a similar radius been used by others? More importantly, it should be clearly stated what this radius is meant to represent - the size of the cyclone or the size of the area which can affect the subsequent evolution of the cyclone?

**Reply:** Thank you for pointing to this. Indeed, the choice of 1000 km is somewhat arbitrary. As you correctly mention there are essentially two length scales involved that are relevant: one represents the overall size of the cyclone itself, and the other the environment affecting the cyclone's deepening and propagation. The distinction between the two is not clearly discussed in the manuscript, but definitely addresses an important point and will therefore be included in the revised version. In fact, from the key question that we discuss in this study (factors affecting deepening and propagation), the circle with a 1000 km radius must capture the environmental factors relevant for the evolution, as mentioned before. Hence, it does not directly relate to the cyclone size. Why 1000 km? We think that this radius is large enough to allow for a physically reasonable displacement between the cyclone center and, e.g., an upper-level forcing feature. On the other hand, it is small enough not to be influenced by too far away features. Compared to other studies, e.g., Campa and Werrnli (2012) who used a radius of 200 km, our choice is substantially larger. However, in agreement with the argument before, Campa and Wernli (2012) intended to study the PV *structure* of the cyclone, whereas we intend to study its forcing environment. Other studies relied on length similar length scales. We will add an explicit note explaining the radius choice in the revised manuscript.

Čampa, J., & Wernli, H. (2012). A PV Perspective on the Vertical Structure of Mature Midlatitude Cyclones in the Northern Hemisphere, *Journal of the Atmospheric Sciences*, *69*(2), 725-740.

2) Line 138. The three geographic boxes. Are the results sensitive to the choice of these areas? The reason for this comment is that these areas are quite large and cover the central and end parts of the main storm tracks regions. Why not consider the start and end of the storm track regions separately as it is well known that cyclones with their genesis in the western North Atlantic differ from those with their genesis in the eastern North Atlantic.

**Reply:** Thank you for the comment. The boxes are indeed based on the storm tracks in the North Atlantic and North Pacific. Additionally, we added the north america box to include lee cyclogenesis cyclones from the Rocky Mountains. Analysing cyclones in the beginning and end regions of the storm tracks separately is a good idea for a follow-up study/analysis. However, we mainly tried to focus on cyclones in the Northern Hemisphere in general with regards to the two forcings used to identify patterns and underlying mechanisms.

3) Section 2.3 / Figure 1. The 12-hour change in SLP. This diagnostic is not clear to me. At first I thought this was the maximum 12 hour deepening rate (the text on line 147 caused this thought) but this cannot be correct given that it has positive and negative values in the histograms in Figure 1. Please can it be clarified what this is – this aspect caused me quite a lot of confusion throughout the manuscript.

**Reply:** Many thanks for this point, which is indeed unclear in the manuscript. In section 3.2 we write: "Figure 3a shows the forcing histogram, similar as in Fig. 1, but this time exclusively for the cyclones' growth phase…". Hence, in contrast to the results in section 3 and 4, here all cyclone phases (growth and decaying) are considered. This results in negative 12-h SLP changes in the cases of weak forcing. We agree with the reviewer that this piece of information is 'hidden' in the text and needs to be clarified. We have now decided that it would be more consistent if Fig.1 is also restricted only to the growth phase (dimensionless time interval 1 to 0) of the cyclone lifecycle. The figure in the manuscript will be replaced by the following one, for which the 2D forcing diagram matches with Fig. 3A and the example histograms for the four forcing categories and for the 12hour SLP change are, of course, also for the growth phase only.

[Figure]

**Replies Figure 8.** As Figure 1 in manuscript but for the intensification phase only.

4) Related to minor point 3, how does the 12-hour change in mean sea level pressure relate to the normalised time presented in lines ~125 – 135? e.g. How does this "real" time relate to normalised time?

In section 2.2 the normalised cyclone time is introduced, and from this time on only these normalised times are discussed in the manuscript. The reviewer is perfectly right that it would also be interesting to relate the phase of the cyclone evolution, as expressed in the normalised time, as a 'real' time, i.e., in hours. The following figure shows the mean 'real' time corresponding to a normalised time. Thereby, we keep the time instance of minium SLP as time 0, both in real and normalised time. For instance, the normalised time -0.75 corresponds in the mean over all cyclone growth phases to a real time of -48 h, i.e., 48 h before reaching the minium SLP.

Of course, the relationship in the figure only represents the mean times and does not reflect the substantial variability due to very different cyclone life cycles.

[Figure]

**Replies Figure 9.** Relation of normalized time with real time.

5) Lines 154-155. Does this statement that none of the distributions is strongly skewed apply to all areas of the phase space e.g. what would the distributions look like for a point in the middle which has a much larger sample size? Potentially, the distributions become more Gaussian in the middle are are most extreme in the corners of the phase space?

[Figure]

**Reply:** Yes, all distributions in the phase space are resembling a normal distribution. It is also true that the distributions become more gaussian with more values, i.e. the area in and around the middle of the phase space. This can be observed especially well in the Figure to the left, where all the shown distributions incorporate a lot of values and show a gaussian curve. The upper corners tend to become less gaussian as a result of having fewer values.

**Replies Figure 10.** 2D forcing histogram with EGR (in day−1) on the x-axis and QGω (in Pas−1) on the y-axis, the mean of the 12-hour ΔSLP distribution within each 2D bin is colored, with darker red colors indicating a stronger cyclone growth. For selected bins in the centre of the histogram (framed in grey, consisting of one bin), the distribution of ΔSLP values is shown.

6) Line 194 – Very minor comment. It is not clear (to me) what is meant by the Greenland Shelf – is this the land / ice mass of Greenland?

**Reply:** Yes, we mean the steep slope of the ice sheet of Greenland.

**Figure comments**

- Figure 2 caption. Can the months analysed be added here? It would help remind a reader that only the extended cold season is analysed. **Reply:** Thank you, we have added the months to the caption.

- Figure 4 and 5. These are presented on a cyclone-centre relative grid in terms of longitude and latitude. How does the 1000 km radius used earlier relate to this lon / lat space? The physical distance in kilometres between 20 degrees of longitude decreases with increasing latitude. Is it valid to assume that 10 degrees in longitude or latitude is approximately 1000 km? If valid, could this assumption be added to the captions?

**Reply:** The lon-late space was set to this size in order to widely capture and display the significant features around the cyclone centre. Given that some of the composite data come from higher latitudes (60-80 degrees), the assumption that 10 degrees is approximately 1000 km is not valid.

- Figure 4 and 5 – related to major points 3 and 4 above. The mean values are presented in these cyclone composites but how much variability is there within each of these composites? This variability may be large due to the different offset times considered all together here. Furthermore, the variability may differ considerably between the 4 classes given the difference in the number of samples in each class.

**Reply:** We agree with the reviewer that the composites do not represent potential variability within the cyclone environment. However, the similarities of the composites in Replies Figure 7 and figures 4 and 5 in the manusctipt indicate that the variability related to separate parts of the intensifiaction phase is low.

- Figure 7. What are the grey / blocked out areas in the bottom left and bottom right of this figure? Also the caption needs a capital letter for "additionally".

**Reply:** These are regions above 1500 m (~850 hPa). The grammar mistake has been corrected.

- Figure S2 has a odd map projection and is lacking longitude and latitude labels. Can this projection / figure style be changed to match the maps shown in the main manuscript?

**Reply:** We agree. Actually, it is a regular lat/lon projection, but the figure is distorted in the y direction. We will redo the figure and apply the same projection/settings as in the main text.

**References**

Catto, J.L., Shaffrey, L.C. and Hodges, K.I., 2010. Can climate models capture the structure of extratropical cyclones?. Journal of Climate, 23(7), pp.1621-1635.

Dacre, H.F., Hawcroft, M.K., Stringer, M.A. and Hodges, K.I., 2012. An extratropical cyclone atlas: A tool for illustrating cyclone structure and evolution characteristics. Bulletin of the American Meteorological Society, 93(10), pp.1497-1502.

Flaounas, E., Raveh-Rubin, S., Wernli, H., Drobinski,  P. and Bastin, S., 2015. The dynamical structure of intense Mediterranean cyclones. Climate Dynamics, 44(9-10), pp.2411-2427.

---

## Referee Report (RR1)

The authors have responded to almost all of my previous comments in a very satisfactory manner. There are two exceptions: details of the ΔSLP diagnostic (#3 below) and the effect of the bin size (#4 below) which in my opinion still require further consideration and revisions to the manuscript. Furthermore, when reading the revised manuscript, I noticed a few minor issues that lacked clarity (points 1, 2, 5, 6) and thus a few additional revisions are still warranted to improve both the clarity of the manuscript.

1. Line 60. This new paragraph in the introduction does not flow well with the rest of the text and it is not clear whether the purpose of this paragraph is the motivate / justify the choice of the two variables or explain what they are. I think this paragraph should be primarily motivation and this needs to be written more clearly. e.g *The main motivation for selecting these two variables is..."*

2. Line 99. This sentence is confusing *"The static stability is not constant in the domain, but a 1d vertical profile is used instead"* as it includes reality (static stability varies in the horizontal and vertical) and what is done to compute to QG omega (static stability is assumed constant in the horizontal and only varies in the vertical). A few more words are needed in this sentence to make it clearer.

3. Figure 1. ΔSLP. The issue I previously raised (see reviewer 2, minor point 3) has not been resolved. It is still unclear when reading the manuscript exactly how this diagnostic is calculated, for example if the real time between time of genesis and time of minimum MSLP is 48 hours, are there four 12-hr values of deepening rate calculated (-48 to -36, -36 to -24, -24 to -12 and -12 to 0hr) or is a sliding window used or is only the maximum value used? Secondly, a sentence needs to be added to the manuscript explaining why there are positive values (weakening cyclones) in the distributions of figure 1 even when only the normalised times from -1 to 0 (the intensification phase) are considered.

4. Previously, I asked how the size of the bins and the differing number of points per bin may affect the results. Thank you for providing additional analysis on this matter. However, having seen Figure 6 in your replies, I do not think your conclusions (first bullet point in conclusions section and also text in section 3.2) as currently written are fully supported by your analysis - I think it is more complex than you state and there are a few subtle points that you should stress more clearly.

   Firstly, Figure 6 in your reply, bottom right panel (zoomed in part) shows that, for this limited part of the parameter space, that the strongest deepening rates occur for high EGR but low QG omega. This is not consistent with Figure 3a in the manuscript and needs explaining. Potentially the mean values in this part of the parameter space are not statistically different though and considering the distributions may clarify this point.

   Secondly, when Figure 6 in your reply is considering together with the 2D histogram shown in Figure 3a in the manuscript, I think the correct interpretation of these figures / analysis is that the while the strongest deepening rates occurs for high EGR and high Q, strong deepening rates can also occur for high EGR and moderate values of QG omega. This is somewhat written in the first bullet point of the conclusions but I feel it is a result which should be stressed more and better explained. Related to this, Figure 3a strongly suggests that EGR has a stronger influence on deepening rate than QG omega – this is already somewhat touched on (but rather indirectly and briefly) by the authors when discussing the asymmetry between the bottom right and top left corners. This subtle result is quite interesting and the manuscript would benefit if this was highlighted more clearly and physically explained. One hypothesis to consider is can the instability (high EGR) be

effectively released as long as there is a reasonable amount of upper level forcing? (moderate to high Q)?

In summary, the authors should carefully revisit and revised section 3.2 and the first bullet point in the conclusions, potentially even splitting this conclusion into two.

5. Line 160. Suggest you revise "49 bins" to "49 linearly distributed bins".

6. Line 276-277. This sentence could be clearer – currently it sounds rather alarming (that the results are almost meaningless). I think the authors intend to say that the mean evolution shown in Figure 4 is not representative of any one individual cyclone lifecycle. Please revise this.

7. Figure 4. While this is a nice addition to the manuscript, showing only the mean values clearly hides the large amount of variability as the values on the y-axes cover much smaller ranges than what is shown on the x- and y-axis of Figure 3. Is it possible to add some additional lines to this figure e.g the 25th and 7th percentile values?

---

## Author Response (AR2)

**Replies to editor comments**

Dear authors,

The two reviewers are satisfied with the way the first revision has been made. But both reviewers have still some suggestions and comments you will have to consider. Like one of the reviewer, I would appreciate a better writing of the new paragraph in the introduction describing the choice of the two variables. Maybe in this paragraph or in the section method (subsection 2.1) it would be worth mentioning the link between the two variables. The temperature horizontal gradient intervenes in the Q-vector definition and in that sense the EGR is involved in the Q-vector definition. For a given upper-level disturbance, the omega-forcing might be stronger depending on the intensity of the baroclinicity. Of course the presence of a synoptic upper-level disturbance renders the two diagnostics different but maybe some sentences relating the two variables could be useful for future readers. I look forward to the new version of the paper.

Best regards,
Gwendal Rivière

Dear Gwendal

We polished the paragraph in the introduction (Line. 60) and added in Section 2.1 a comment on the link between the two variables (via the horizontal temperature gradient) and that they are not fully independent. In our study the EGR is computed at lower levels, while the Q-vector is computed at upper levels. In our case it is therefore not the same temperature gradient that enters the computation of both variables. But in general, we agree that this would be the case if the variables were computed on the same vertical level.

Best regards
The authors

**Replies to comments by Reviewer 2**

The authors have responded to almost all of my previous comments in a very satisfactory manner. There are two exceptions: details of the ΔSLP diagnostic (#3 below) and the effect of the bin size (#4 below) which in my opinion still require further consideration and revisions to the manuscript. Furthermore, when reading the revised manuscript, I noticed a few minor issues that lacked clarity (points 1, 2, 5, 6) and thus a few additional revisions are still warranted to improve both the clarity of the manuscript.

1. Line 60. This new paragraph in the introduction does not flow well with the rest of the text and it is not clear whether the purpose of this paragraph is the motivate / justify the choice of the two variables or explain what they are. I think this paragraph should be primarily motivation and this needs to be written more clearly. e.g "The main motivation for selecting these two variables is..."
The paragraph starting a Line 60 was polished and now contains our main motivation for the choice of the two variables. Our motivation results from the classical picture of baroclinic

cyclone development. As suggested by the reviewer we now start the paragraph explicitly with *"The main motivation for the selection of these two variables results from the classical picture of the extra-tropical cyclone development."* We also streamlined the remainder of the paragraph to make the choice of our forcing factors, hopefully, very clear to the readers.

2. Line 99. This sentence is confusing "The static stability is not constant in the domain, but a 1d vertical profile is used instead" as it includes reality (static stability varies in the horizontal and vertical) and what is done to compute to QG omega (static stability is assumed constant in the horizontal and only varies in the vertical). A few more words are needed in this sentence to make it clearer.

We added a few more sentences in Line 103. The need to use a 1d vertical profile arises from numerical problems in the inversion, which might occur in situations of near neutral or negative static stability. Instead of using the full 3d static stability, a horizontal average is used to compute a 1d reference profile at each time step. This pragmatic choice has been made already in previous studies and does not substantially affect the final outcome.

3. Figure 1. ΔSLP. The issue I previously raised (see reviewer 2, minor point 3) has not been resolved. It is still unclear when reading the manuscript exactly how this diagnostic is calculated, for example if the real time between time of genesis and time of minimum MSLP is 48 hours, are there four 12-hr values of deepening rate calculated (-48 to -36, -36 to -24, -24 to -12 and -12 to 0hr) or is a sliding window used or is only the maximum value used? Secondly, a sentence needs to be added to the manuscript explaining why there are positive values (weakening cyclones) in the distributions of figure 1 even when only the normalised times from -1 to 0 (the intensification phase) are considered.

We clarified this in Section 2.3 near L. 174. We do not use a sliding window or the maximum value, instead we use all time intervals between genesis and minimum MSLP. In your example of a 48 hour growth period there are four values that contribute to the statistic. For this reason positive values occur when the deepening phase is not a simple linear deepening and are accepted because we consider the full growth period.

4. Previously, I asked how the size of the bins and the differing number of points per bin may affect the results. Thank you for providing additional analysis on this matter. However, having seen Figure 6 in your replies, I do not think your conclusions (first bullet point in conclusions section and also text in section 3.2) as currently written are fully supported by your analysis - I think it is more complex than you state and there are a few subtle points that you should stress more clearly.
Firstly, Figure 6 in your reply, bottom right panel (zoomed in part) shows that, for this limited part of the parameter space, that the strongest deepening rates occur for high EGR but low QG omega. This is not consistent with Figure 3a in the manuscript and needs explaining. Potentially the mean values in this part of the parameter space are not statistically different though and considering the distributions may clarify this point.

Yes indeed. We must not expect a linear increase in the deepening rates with linearly increasing EGR and QG Omega. The standard deviation was added to Figure S6 of our manuscript for each bin to clarify this issue. Because the standard deviation in each bin is larger than the difference between two consecutive mean values between two bins, we must expect that due to a certain skewness of the distribution that very high values occur in a bin that has however a lower mean value compared to its neighbouring bin.

Secondly, when Figure 6 in your reply is considering together with the 2D histogram shownin Figure 3a in the manuscript, I think the correct interpretation of these figures / analysis is that the while the strongest deepening rates occurs for high EGR and high Q, strong deepening rates can also occur for high EGR and moderate values of QG omega. This is somewhat written in the first bullet point of the conclusions but I feel it is a result which should be stressed more and better explained. Related to this, Figure 3a strongly suggests that EGR has a stronger influence on deepening rate than QG omega – this is already somewhat touched on (but rather indirectly and briefly) by the authors when discussing the asymmetry between the bottom right and top left corners. This subtle result is quite interesting and the manuscript would benefit if this was highlighted more clearly and physically explained. One hypothesis to consider is can the instability (high EGR) be effectively released as long as there is a reasonable amount of upper level forcing? (moderate to high Q)? In summary, the authors should carefully revisit and revised section 3.2 and the first bullet point in the conclusions, potentially even splitting this conclusion into two.

*This is a very interesting comment and suggestion. Thank you! A detailed analysis of the asymmetry between EGR and QGω forcing would be interesting, but we also think that it asks for a systematic and detailed analysis, and hence goes beyond the scope of this study. Still, we are happy to include the reviewer's comment into the manuscript. More specifically, we have added the following text to section 3.2 (closely following the reviewer's suggestion):*

> *The asymmetry between the two opposite corners could also point to a (potentiallly) subtle difference in EGR and QGω forcing. It indicates that EGR has a stronger influence on the deepening rates than QGω, and that baroclinic instability might be released as long as there is a reasonable (even moderate) amount of upper-level forcing. In short, moderate upper-level forcing (QGω) might be sufficient to trigger substantial deepening rates if EGR is high. In contrast, if EGR is low, weaker deepening rates result even if substantial upper-level forcing is discernible.*

*Additionally, as suggested by the reviewer, we discuss this specific point in an extra bullet point in the conclusions.*

*An interesting asymmetry is discernible between conditions with high EGR and low QGω forcings and, conversely, low EGR and high QGω forcing: larger deepening rates result for the former than the latter. This indicates that baroclinic instability and substantial deepening rates can result even in situations with moderate upper-level (QGω) forcing as long as the EGR is high enough. The opposite is, however, not true: even substantial upper-level forcing results only in weak deepening rates if EGR is low.*

5. Line 160. Suggest you revise "49 bins" to "49 linearly distributed bins".
*Corrected.*

6. Line 276-277. This sentence could be clearer – currently it sounds rather alarming (that the results are almost meaningless). I think the authors intend to say that the mean evolution shown in Figure 4 is not representative of any one individual cyclone lifecycle. Please revise this.
*The sentence was revised accordingly (now L. 286).*

7. Figure 4. While this is a nice addition to the manuscript, showing only the mean values clearly hides the large amount of variability as the values on the y-axes cover much smaller ranges than what is shown on the x- and y-axis of Figure 3. Is it possible to add some additional lines to this figure e.g the 25th and 7th percentile values?
The Figure has been updated and includes the mean values + 0.5 STD in either direction.

**Replies to comments by Reviewer 1**

The authors have satisfactorily addressed all my comments and concerns and I think the paper is suitable for publications after some minor revision. I only have a few comments:

1. I still think it's worth adding the separate Q and E maps (Fig. 1 in the response) to the SI, since it does provide additional information. It shows for each one of the cases Q(up,down) E(up,down) where the main contribution is coming from (from either Q,E, or from both), which is interesting.

The separate Q and E maps have been added to the Supplement.

2.Fig.2a in the paper- is it possible that there is some mistake in the calculation here? Why is the density distribution almost three times larger than in the other cases? Especially after seeing Fig.1 in the response, this is not clear to me, since the region of Q_up and E_down do not overlap much. For example, based on Fig.1 in the response, I would expect Q_up and E_up (whose regions overlap) to have higher density than the former.

It seems odd that the Q_up E_down density is so much higher than the other ones. As you stated, it feels counterintuitive when considering the separate Q and E distributions. However, one thing to note is that when calculating the density for the separate distributions, we use a different data range. For instance: For E_down, we use only the lower range of EGR but use the entire range of QGomega (in the Forcing Histogram, this corresponds to the first two columns and all rows). However, for Q_up E_down, we only use the first two columns and first two rows (upper left corner of the histogram). This difference in data range may account for discrepancies between the combined forcing distributions and the separate distributions for E and Q.

3. Can this study be useful for understanding why the midlatitude storm tracks are too zonal in global simulations (e.g., in historical CMIP simulations)?
Yes, thank you for pointing to this potential application of the method. We added the following final statement at the end of the paper.

> *Finally, the combined EGR and QGω perspective could be used to identify biases in the representation of cyclone dynamics and storm tracks in present-day and historic CMIP6 simulations (Priestley et al., 2020). For instance, midlatitude storm tracks are known to be too zonal in historical CMIP6 simulations, and the robust EGR/QGω diagnostic could readily be applied to temporally coarser dataset than the six-hourly ones used in this study.*